# Spatial patterns of hepatocyte glucose flux revealed by stable isotope tracing and multi-scale microscopy

Aliyah Habashy[1,11], Christopher Acree [1,11], Keun-Young Kim[2], Ali Zahraei[3,4], Martin Dufresne[3,4], Sebastien Phan[2], Melanie Cutler[1], Emilee Patterson[1], Alexandra G. Mulligan [3], Kristopher Burkewitz [3], Charles Robert Flynn[5], Louise Lantier [1], Thomas Deerinck[2], Owen P. McGuinness [1], Jeffrey M. Spraggins [3,4,6,7,8], Mark H. Ellisman [2] & Rafael Arrojo e Drigo [1,9,10] ✉

Metabolic homeostasis requires engagement of catabolic and anabolic pathways consuming nutrients that generate and consume energy and biomass. Our current understanding of cell homeostasis and metabolism, including how cells utilize nutrients, comes largely from tissue and cell models analyzed after fractionation, and that fail to reveal the spatial characteristics of cell metabolism, and how these aspects relate to the location of cells and organelles within tissue microenvironments. Here we show the application of multi-scale microscopy, machine learning-based image segmentation, and spatial analysis tools to quantitatively map the fate of nutrient-derived [13]C atoms across spatiotemporal scales. This approach reveals the cellular and organellar features underlying the spatial pattern of glucose [13]C flux in hepatocytes in situ, including the timeline of mitochondria-ER contact dynamics in response to changes in blood glucose levels, and the discovery of the ultrastructural relationship between glycogenesis and lipid droplets.

Tissue function is supported by cell metabolism pathways that are modulated to meet changes in nutrient availability and energetic demands that occur throughout an organism's lifetime. Much of our knowledge regarding cell metabolism is derived from bulk metabolomics using stable or radioactive isotopes (i.e., [13]C and [14]C, respectively). Throughout the years, this approach has revealed differences in how cells utilize nutrients to maintain energy and cell homeostasis during different cell states[1,2], including how these aspects are impacted by aging, cancer, and degenerative and metabolic diseases[3]. Cells are organized into sub-cellular compartments created by organelles that handle essential processes necessary for cell function, such as mitochondrial respiration or protein synthesis within the endoplasmic reticulum (ER)[4,5]. Several aspects of cell metabolism require proper organization of organelle-interaction networks that create distinct intracellular compartments such as mitochondria-ER or mitochondria-lipid droplet contact sites[6,7]. These sub-cellular compartments are

[1]Department of Molecular Physiology and Biophysics, Vanderbilt University, Nashville, TN, USA. [2]National Center for Imaging and Microscopy Research (NCMIR) and the Department of Neurosciences, University of California San Diego, School of Medicine, La Jolla, CA, USA. [3]Department of Cell and Developmental Biology, Vanderbilt University, Nashville, TN, USA. [4]Mass Spectrometry Research Center, Vanderbilt University School of Medicine, Nashville, TN, USA. [5]Department of Surgery, Vanderbilt University Medical Center, Nashville, TN, USA. [6]Department of Biochemistry, Vanderbilt University, Nashville, TN, USA. [7]Department of Chemistry, Vanderbilt University, Nashville, TN, USA. [8]Department of Pathology, Microbiology and Immunology, Vanderbilt University Medical Center, Nashville, TN, USA. [9]Center for Computational Systems Biology, Vanderbilt University, Nashville, TN, USA. [10]Diabetes Research and Training Center (DRTC), Vanderbilt University Medical Center, Nashville, TN, USA. [11]These authors contributed equally: Aliyah Habashy, Christopher Acree. ✉e-mail: r.drigo@vanderbilt.edu

dynamic and interact via proteins that mediate membrane anchoring and the exchange of molecules and ions between organelles[6–9]. Perturbation of these organelle contact sites disrupts cell and whole-body metabolism and has been linked to the patho-physiology of metabolic and neurodegenerative diseases[8,10–13]. Therefore, there is a need to study and understand the principles that guide the spatial organization pattern of cells and organelles in situ and their correlation to changes in animal and cell metabolism.

Different super resolution light and electron microscopy techniques have been applied to determine the architecture and spatiotemporal dynamics of organelle-interaction networks with nanometer resolution, as well as cell-type specific patterns of organelle morphology and connectivity linked to the regulation of cell and whole-organism function and metabolism[10,12,14–20]. Moreover, recent advances in imaging metabolomics techniques such as MALDI-MS[21,22] and ToF-SIMS[23] have allowed the visualization of the spatial distribution of metabolites and molecular flux at tissue and multi-cellular scales. However, due to their intrinsic working physics, these techniques are unable to measure the fate of metabolites at sub-cellular resolution and therefore are unable to determine the correlation between nutrient flux and underlying cell and organelle organization patterns.

In recent years, we have developed a correlative microscopy pipeline that combines scanning electron microscopy (SEM) with multi-isotope mass spectroscopy (MIMS) called MIMS-EM[24]. MIMS-EM leverages SEM's high spatial resolution and the high-resolution mass detectors of MIMS to map stable isotope incorporation (e.g., $^{15}N$, $^{14}N$, $^{13}C$, or $^{12}C$) into macromolecules and organelles, creating spatial maps of isotope flux and (intra)cellular architecture. We previously used MIMS-EM and stable isotope-labeling of whole animals and cells to identify the vast heterogeneity of protein and cell longevity[24–26]. Importantly, MIMS-EM was based on landmark studies that established MIMS as a framework to follow cell and protein turnover, nutrient-synthesis, and elemental flux in relatively short time scales[27–30].

In this work, we demonstrate the application of our multi-platform approach to quantify glucose flux and determine the fate of glucose-derived elements – from whole animals to subcellular compartments in situ. By combining orthogonal and complimentary experimental and high-resolution microscopy techniques with machine-learning (ML) data analysis and stable isotope labeling of mice (SILAM) using [U-$^{13}C_6$]-glucose tracers, we were able to determine the spatial and the molecular flux patterns of glucose metabolism at whole animal, tissue, and cellular scales. This approach revealed sub-cellular organization patterns that identified organelle-organelle networks and subcellular scaffolds associated with glucose metabolism, and how these patterns are modulated by changes in metabolic demand such as fasting and initiation of glycogen synthesis. This work establishes a multi-modal framework to study the multiple scales of metabolism and the spatial landscape of cells and nutrient flux in situ.

## Results and Discussion
### In vivo labeling of mice with [U-$^{13}C_6$]-glucose to quantify whole-body glucose metabolism
To measure the flux of glucose from whole body to the organelle level and its correlation with cellular and organelle organization, we created a multi-modal pipeline combining the delivery of [U-$^{13}C_6$]-glucose with both traditional mass spectrometry techniques and MIMS-EM (Fig. 1A). We started by delivering [U-$^{13}C_6$]-glucose to freely moving and awake animals using intra-venous catheters with ports for arterial blood sampling to quantify blood glucose and plasma metabolite $^{13}C$ enrichment (Fig. 1A). Here, fasted 8-week-old male C57/BL6J mice were placed inside individual metabolic cages and continuously infused with 15 or 40 mg·kg$^{-1}$·min$^{-1}$ of [U-$^{13}C_6$]-glucose for up to 4 hours. These doses were chosen to evaluate in vivo glucose metabolism rates in response to glucose dosages that either matched or exceeded the rate of endogenous glucose production in mice[31]. Mice infused with

15 mg·kg$^{-1}$·min$^{-1}$ remained normoglycemic, while mice dosed with 40 mg·kg$^{-1}$·min$^{-1}$ experienced sustained hyperglycemia (Fig. 1B). Next, to investigate the kinetics of whole body [U-$^{13}C_6$]-glucose oxidation in vivo, we measured the relative enrichment of $^{13}C$ in the expelled breath $CO_2$ using stable isotope mass spectrometer gas detectors coupled to our metabolic cages (Fig. 1C, and Supplementary Fig. 1A). This approach quantified time- and dose-dependent increases in $^{13}CO_2$ in [U-$^{13}C_6$]-glucose-infused mice, thus confirming that [U-$^{13}C_6$]-glucose molecules were delivered and oxidized within the first 60 minutes (Fig. 1C). Accordingly, exchange of [U-$^{13}C_6$]-glucose for unenriched glucose caused $^{13}CO_2$ to quickly fall over time (Fig. 1C). Similar results in mice exposed to a longer fast (i.e., 16 hours) and infused with 40 mg·kg$^{-1}$·min$^{-1}$ of [U-$^{13}C_6$]-glucose (Supplementary Fig. 1B–D), validated our stable isotope delivery and quantification of [U-$^{13}C_6$]-glucose oxidation rates and homeostasis in vivo.

In response to an increase in blood glucose, pancreatic beta cells secrete insulin to normalize blood glucose levels[32]. Insulin acts on skeletal muscle depots that metabolize glucose into secondary metabolites that can be measured in the circulation (i.e., lactate and pyruvate), and stimulates the liver and adipose tissue to store glucose-derived carbons into large macromolecules such as glycogen or triglycerides, respectively. To investigate the amount of [U-$^{13}C_6$]-glucose and $^{13}C$-labeled glucose-derived circulating metabolites, we performed GC-MS on plasma samples collected during our infusion experiments. This identified a gradual and significant decrease in the fractional abundance of plasma $^{12}C_6$-glucose (M + 0) and an increase in plasma $^{13}C$-labeled glucose (M + 6) (Fig. 1D). A similar pattern in the appearance of M + 3 was also observed for several circulating metabolites such as lactate, pyruvate, glycerol, and alanine (Fig. 1E, Supplementary Fig. 1E–G, and Supplementary Table 1). Finally, we quantified $^{13}C$ enrichment in newly synthesized glycogen molecules in the liver using GC-MS and found a time-dependent increase in $^{13}C$-labeled glycogen molecules, as expected (Supplementary Fig. 1H). Together, these results indicate that as [U-$^{13}C_6$]-glucose floods the circulatory system, the relative abundance of glucose-$^{13}C$ in liver glycogen and in plasma glucose metabolite pools increases over time.

### Quantification of glucose-derived $^{13}C$ incorporation at the tissue level
Hepatocytes are organized in distinct architectural zones with unique transcriptional, metabolic, and organelle organization profiles that underlie differences in glucose metabolism and glycogenesis[19,33–35]. To explore how the multiple layers of hepatocyte organization impact glucose metabolism in situ, we developed a microscopy-based approach to investigate specific aspects of glucose-$^{13}C$ flux in space. First, to quantify liver glycogen synthesis and $^{13}C$ enrichment in situ, we applied MALDI-MS to snap-frozen livers from mice infused for 4 hours with 40 mg·kg$^{-1}$·min$^{-1}$ [U-$^{13}C_6$]-glucose after an overnight fast (Fig. 2A, Supplementary Fig. 2A). We chose these conditions because we wanted to study $^{13}C$ flux in animals experiencing a large shift in metabolism marked by increased demands for glycogen synthesis and glucose handling[36]. Our MALDI-MS approach was based on a previously established method using isoamylase to hydrolyze glycogen alpha-1-6-glycosidic linkages to measure glucose chain abundance[37] (Fig. 2B). This glycogen MALDI-MS imaging approach achieved a spatial X-Y resolution of ~20um and revealed that 4-hour $^{13}C$-glucose-infused livers had ~8x more glycogen than a fasted liver section (Fig. 2C and Supplementary Fig. 2B). Moreover, imaged glycogen depots had an overall fractional $^{13}C$ enrichment of ~30-50%, which is compatible with our GC-MS data (Supplementary Fig. 1H). We also determined the molecular pattern of $^{13}C$-glycogen enrichment via glycogen MS/MS and identified the spectra of a total of fifteen different glucose polymers (GP). Identified GPs ranged from 3-to-18 linked glucose molecules (GP3 to GP18), and most GPs were significantly enriched with $^{13}C$ (e.g., GP7 with three $^{13}C_6$-glucose molecules ($^{13}Cx3$), Supplementary Fig. 2C);

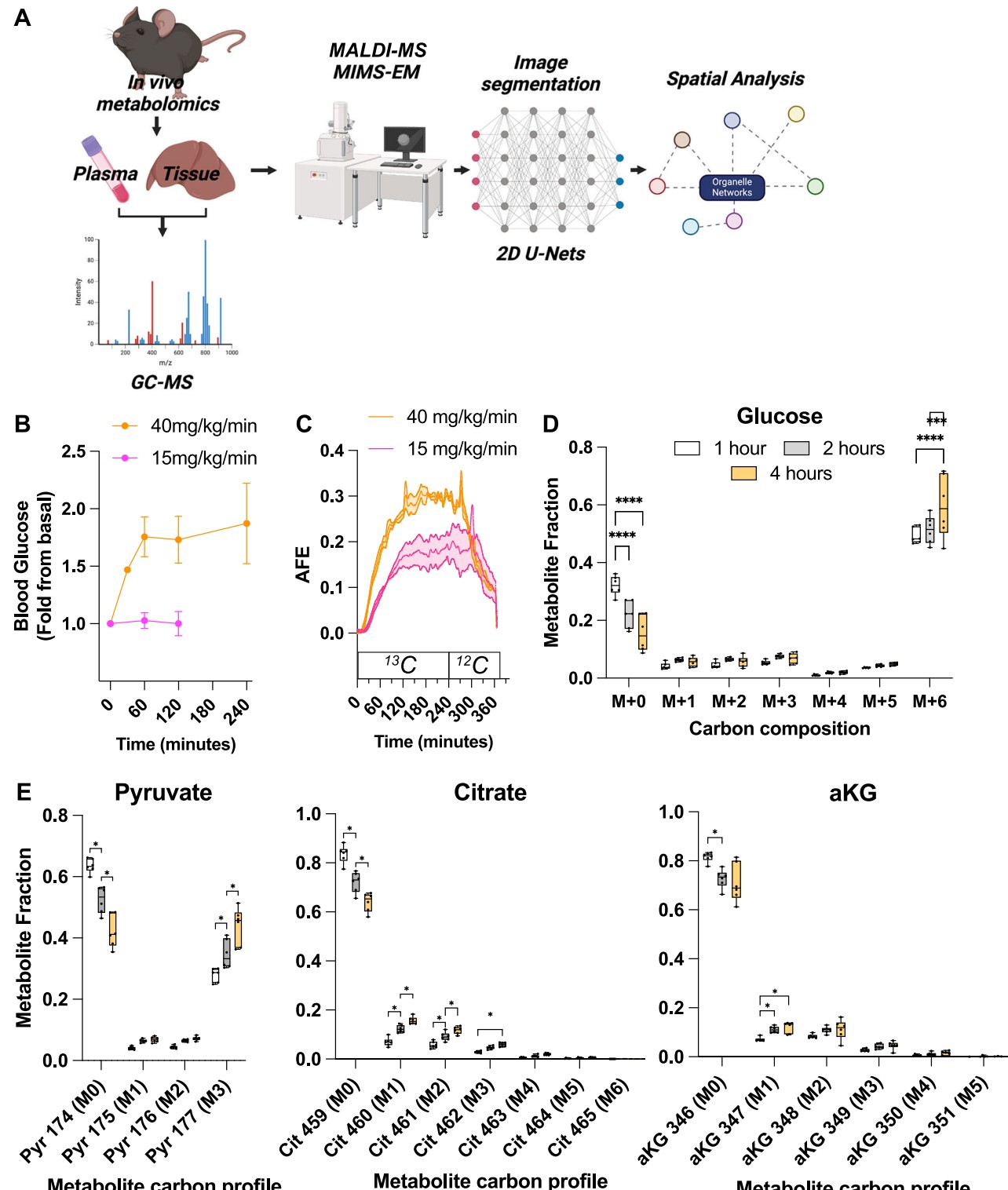

**Fig. 1 | In vivo labeling of mice using [U-$^{13}$C]-glucose. A** Illustration of the approach used to label freely moving and unanesthetized mice with [U-$^{13}$C]-glucose for up to 4 hours using catheters followed by MIMS-EM and spatial analysis. Figure created using biorender.com. **B** Blood glucose levels measured from mice continuously infused with 15 or 40 mg/min/kg of total body mass for up to 4 hours. Data from $n$ = 4-17 mice per group. **C** Expelled $^{13}$CO$_2$ (in parts per million (ppm)) measured from the atmosphere of custom-made metabolic cages using gas mass spectrometers. $^{13}$C glucose was infused for the first 240 minutes and replaced with $^{12}$C glucose for an additional 120 minutes. AFE, atomic fractional enrichment. Data from $n$ = 3 mice per group. **D, E** GC-MS analysis to determine the fractional $^{13}$C enrichment of circulating glucose molecules and of secondary metabolites pyruvate, citrate, and alpha ketoglutarate (aKG) generated from glucose metabolism. Each dot represents an animal, $n$ = 6 animals per time point. In **B**, data shown as ± standard deviation of the mean. In **C**, the shaded region indicates the data range of $^{13}$CO$_2$ measurements. In **D, E**, ****$p < 0.001$ and *$p < 0.05$ using One-way ANOVA with Kruskal-Wallis tests.

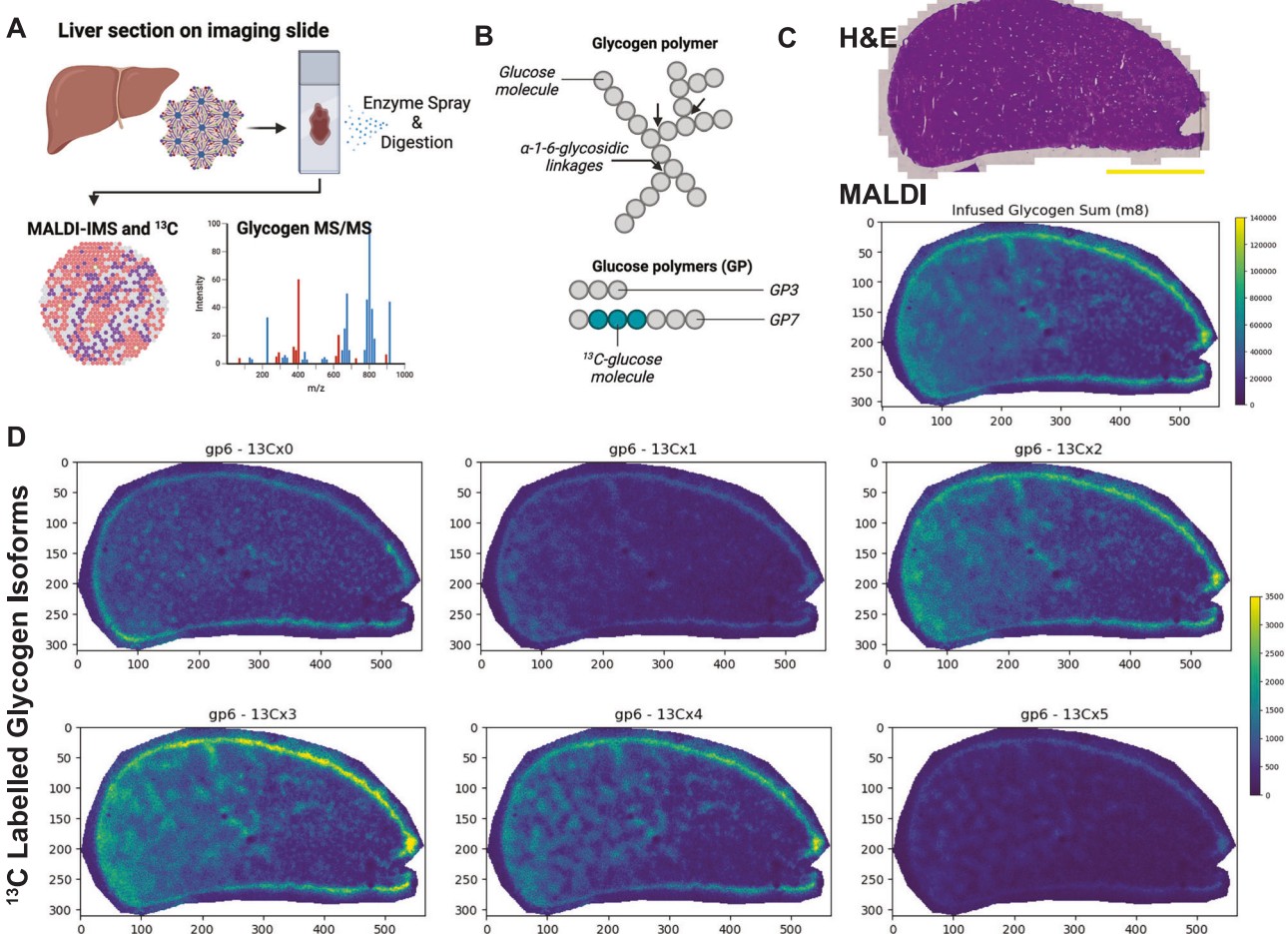

**Fig. 2 | MALDI-MS imaging of ¹³C-labeled glycogen in mouse liver tissue.**
**A** Schematic of sample processing and MALDI mass spectrometry (MS) imaging of mouse livers from control and ¹³C-labeled mice. **B** Cartoon illustration of a mature glycogen molecule (top) where alpha-1-6-glycosidic linkages are shown. These linkages are targeted by isoamylase digestion to release glucose polymers (GP) of different sizes (below). Grey and teal dots represent ¹²C and ¹³C-labeled glucose molecules, respectively. **C** Representative images of hematoxylin and eosin (H&E) staining (top), and total glycogen content map measured using MALDI-MS of a mouse liver section after 4 hours of [U-¹³C₆]-glucose infusion. Scale bar, 1 mm. **D** Same as in **C**, however MALDI-MS images show the spatial pattern of a specific ¹²C- and ¹³C-labeled glucose polymer (GP) with six glucose molecules (GP6). In each panel, the number after "¹³Cx" indicates the number of ¹³C-labeled molecules found in GP6. In **A**–**D**), data representative of $n = 3$ mice. **A**, **B** created using biorender.com.

however, significant amounts of unenriched (¹²C)-glycogen were also detected (e.g., GP6 ¹³Cx0) (Fig. 2D). Notably, we applied "untargeted" MALDI-MS to identify other potential ¹³C-labeled molecules, however, we only identified ¹³C₆-glucose and ¹³C₆-hexose-6-phosphate (H6P) (Supplementary Fig. 2D, E). No specific tissue distribution pattern in glycogen or GPs was identified, likely due to the long and sustained phase of elevated glucose-infusion that made difficult to separate the expected zone differences in glycogen synthesis. Nevertheless, our results support the application of MALDI-MS to measure glycogen ¹³C in situ at the tissue level.

## Spatial enrichment of glucose ¹³C at cellular and subcellular scales

To determine the spatial fate and quantify the flux of ¹³C at the single-cell and subcellular levels in hepatocytes, we applied MIMS-EM. We focused our MIMS-EM imaging sessions to hepatocytes close to the central vein because of their higher glycogenic potential[34]. We used MIMS-EM to collect data for multiple isotopes (i.e., ¹³C, ¹²C, ³²S, and ¹⁴N) to guide image co-registration to hepatocyte SEM micrographs (Supplementary Fig. 3A–E). MIMS-EM of hepatocytes revealed both time- and dose-dependent accumulations of glucose-derived ¹³C within the total hepatocyte biomass following [U-¹³C₆]-glucose infusion

(Fig. 3A–F, Supplementary Fig. 4A). Of note, since delivery of 15 mg·kg⁻¹·min⁻¹ [U-¹³C₆]-glucose closely matches the endogenous mouse glucose disposal rate, hepatocytes engage in low levels of glycogenesis and thus explain the low levels of ¹³C enrichment observed (Fig. 3F).

The spatial enrichment and distribution pattern of hepatocyte ¹³C was granular and largely co-localized with cytosolic glycogen stores (seen in the SEM micrographs as electron dense clusters[38]) that grew larger over time, thus indicating that these depots contained newly synthesized glycogen molecules. Recent studies have established that changes in organelle architecture and organelle interaction networks can affect several aspects of cell function and whole-body metabolism[6,10,13,39]. Therefore, to create a comprehensive map of organelle organization and ¹³C enrichment in subcellular compartments, we created a computational framework to map the spatial organization of individual organelles and to quantify organelle-specific ¹³C enrichment. This was achieved by training 2D U-nets to segment hepatocyte mitochondria, LDs, ER, and glycogen compartments (Fig. 3G, H). These organelle segmentation algorithms were benchmarked against a representative subset of manually annotated SEM images to create organelle classifiers with at least 90% confidence and a < 5% false positive organelle identification rate (Supplementary Fig. 5A–H).

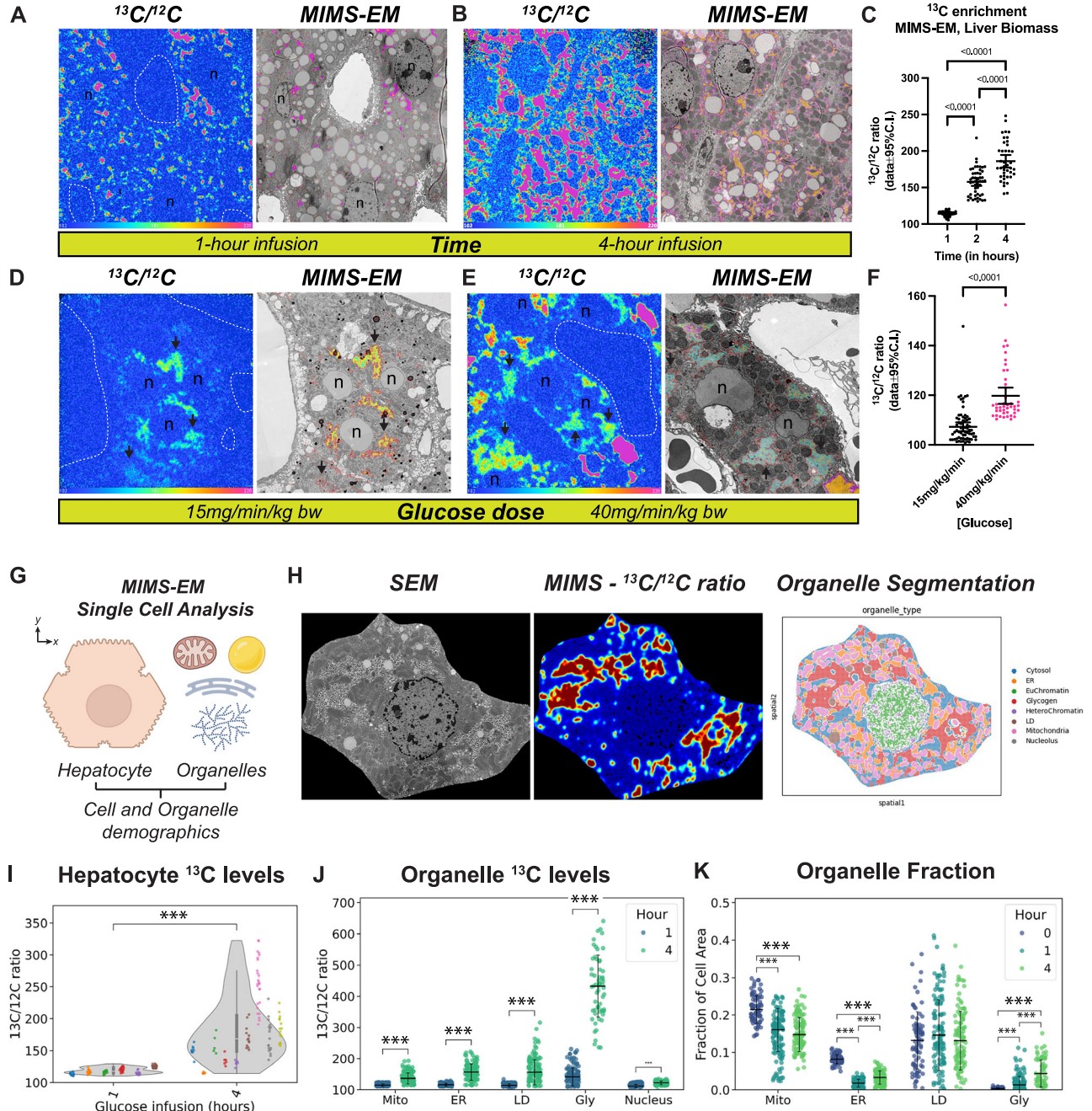

**Fig. 3 | Quantification of liver tissue ¹³C levels using MIMS-EM. A, B** Correlated ¹³C-to-¹²C (¹³C/¹²C) ratiometric images acquired using MIMS and registered to scanning electron microscopy (SEM) of hepatocytes to create MIMS-EM maps. Data from mice continuously infused with [U-¹³C]-glucose at 40 mg/min/kg of [U-¹³C]-glucose for 1 or 4 hours. **C** Quantification of ¹³C/¹²C ratios in the biomass of mice shown in **A, B**. Data from n = 3 animals per time point. Each dot represents the mean ¹³C enrichment in a ~40 um² liver section. **D, E** MIMS ¹³C/¹²C ratiometric images registered to hepatocyte SEM micrographs to create MIMS-EM maps. Data from mice continuously infused with 15 or 40 mg/min/kg of total body mass for 2 hours. **F** Quantification of ¹³C/¹²C ratios in the biomass of mice from shown in **D, E**. In **A, B** and **D, E**, magenta and orange colors represent lower and higher different levels of ¹³C enrichment, respectively. Dotted magenta line marks the terrestrial background for ¹³C. Data from n = 3 animals per time point. Each dot represents the mean ¹³C enrichment in a ~40 um² liver section. **G** Cartoon illustration showing the single cell analysis of MIMS-EM data. Figure created using biorender.com.

**H** Representative hepatocyte SEM, ¹³C/¹²C ratio, and overlay of several organelle segmentation masks, listed on the right. Data from a mouse continuously infused with 40 mg/min/kg of [U-¹³C]-glucose for 4 hours. **I** ¹³C/¹²C ratios of individual hepatocytes from mice continuously infused with 40 mg/min/kg of [U-¹³C]-glucose for 1 or 4 hours. Each color represents data from a different animal. **J** ¹³C/¹²C levels by type of organelle after 1 or 4 hours of 40 mg/min/kg [U-¹³C]-glucose infusion.
**K** Relative fraction of hepatocyte cell area occupied by endoplasmic reticulum (ER), lipid droplets (LDs), glycogen, mitochondria, or nucleus. Data from n = 3-to-7 mice per group, from overnight fasted mice or from mice continuously infused with 40 mg/min/kg of [U-¹³C]-glucose for 1 or 4 hours. In **I, J**, data from n = 6-to-9 mice per group. In **C** and **F**), a One-way ANOVA with a two-stage Benjamini, Krieger, and Yekuteli test was used, and p-values < 0.001 are shown. In **J, K**), One-way ANOVA with Dunns test post-hoc where ***p < 0.001. In **I−K**, each dot represents a single cell. In (**C** and **F-K**), data shown as 95% of the confidence interval (C.I.). In (**A**), scale bar 2 microns; in **D** and **H**, 5 microns.

We applied this approach to measure organelle composition and [13]C-enrichment levels at cell and organelle scales after an overnight fast or 1 or 4 hours of 40 mg·kg$^{-1}$·min$^{-1}$ [U-[13]C$_6$]-glucose infusion. Hepatocytes from [13]C-infused animals had significant [13]C enrichment at the single cell level and in all major organelle compartments identified (Fig. 3I, J, Supplementary Fig. 4B, C). Glycogen depots had the highest levels of enrichment, followed by ER, LD, mitochondria, and other cytosolic compartments. This metabolic signature correlated with a loss of cytosolic area occupied by mitochondria and ER while glycogen grew significantly (Fig. 3K, Supplementary Fig. 4D). Loss of mitochondria density could be explained by increases in mitochondria circularity and/or fragmentation that occurs during re-feeding, whereas the ER becomes compressed into dense stacks. The transient increase and then loss of LD area can be explained by a suppression of lipolysis mediated by rising insulin signaling followed by a decrease in liver fatty acid delivery with relative maintenance in hepatocyte triglycerides secretion via VLDL particles[40].

To place these results in a tissue- and cell-type-specific context, we applied MIMS-EM to monitor glucose-[13]C flux in adult brown adipocytes of mice studies under mild cold stress conditions (i.e., 23 °C room temperature). During brown adipocyte development and adult tissue function, glycogen metabolism is important for the formation of lipid droplets (LDs) and thermogenesis[41,42]. Brown adipocytes metabolize glucose via *de-novo* lipogenesis to sustain the synthesis of small multilocular LDs that interact with a dense mitochondrial population engaged in oxidative and glycolytic glucose metabolism pathways to generate energy and replenish LD content[43]. Accordingly, MIMS-EM of brown adipocytes from 4-hour 40 mg·kg$^{-1}$·min$^{-1}$ [U-[13]C$_6$]-glucose-infused mice revealed significant enrichment of [13]C in LDs, and little-to-no enrichment in cytosolic, mitochondrial, or nuclear regions (Supplementary Fig. 4E). Here, we did not identify glycogen clusters near LDs, which is likely due to the depletion of existing glycogen stores by overnight fasting or a preferential shunt of glucose [13]C metabolism towards *denovo* lipogenesis.

Together, this data demonstrates how in vivo metabolic tracing and MIMS-EM can be combined to quantify glucose flux at cell and subcellular scales.

## Glycogenesis occurs near hepatocyte LD scaffolds

While analyzing our MIMS-EM data, we observed a spatial association between LDs and nascent glycogen molecules (Supplementary Fig. 6A). Previous studies have established the spatial relationship between glycogen depots and the endoplasmic reticulum (ER)[44,45]. This could explain how ER-resident enzymes required for glycogen metabolism (Protein Phosphatase 1 (PP1), which activates glycogen synthase (*Gys2*)) contribute to glycogen homeostasis[44,46]. Therefore, we hypothesized that LDs provide a physical scaffold for glycogen synthesis by forming connections with the ER. To further determine the spatial context of glycogenesis in situ, we first analyzed SEM micrographs of mouse pericentral hepatocytes directly after an overnight fast and/or after a 1- or 4-hour [13]C-glucose infusion. We found that most fasted hepatocytes lack glycogen depots, and the periphery of LDs was often occupied by mitochondria (Fig. 4A) – a sign of active fatty acid oxidation[15]. Strikingly, rare fasting hepatocyte LDs had small clusters of glycogen nearby and in direct contact (Supplementary Fig. 6B). Similarly, [13]C-glucose-infused mice had newly synthesized glycogen stores in direct contact and within the immediate neighborhood of LDs within 1 hour; by 4 hours, LDs were largely surrounded by [13]C glycogen (Fig. 4A).

To gain insights into the potential interaction and cytoplasmic localization of glycogenesis enzymes[45], we performed meta-analysis of published proteomic studies from bulk hepatocytes[47], glycogen[48], LDs[49], or mitochondrial fractions[50]. We used the STRING-DB[51] to query protein-protein interactions (PPIs) and reconstruct PPI networks of glycogen-associated proteins, followed by STRING functional enrichment and network clustering (Supplementary Fig. 7A). This discovered a list of proteins mostly linked to ER structure-function, carbon and glycogen metabolism, and fatty acid metabolism (Supplementary Fig. 7B-D). We ruled out potential contamination from other cellular fractions by overlapping protein hits shared between glycogen, LD, and mitochondria (Supplementary Fig. 7E). Importantly, the glycogen proteome included key glycogenesis enzymes (i.e., *Pgm1*, *Ugp2*, *Gys2*, *Gbe1*), as well as proteins involved in glycogenolysis (i.e., *Epm2a/Laforin*, *Agl*, *Pygl*) and glycophagy (i.e., *Gaa* and *Stbd1*) (Supplementary Fig. 7F). ER-resident proteins glycogen synthase-activating *Ppp1ca* and the glycogen phosphorylase *Pygl*[45], were enriched in the glycogen dataset (Supplementary Fig. 7D). These results suggest that glycogen synthesis could occur within a subcellular compartment that clusters enzymes needed for glycogen synthesis and degradation.

To investigate this hypothesis and define the subcellular architecture of the LD-glycogen interaction during the early phases of glycogenesis, we performed electron tomography (eTomo) of hepatocytes from 1-hour [13]C-glucose-infused mice. Glycogen clusters were observed in direct contact with LDs and occupying the space directly adjacent to LDs (Fig. 4B). This subcellular space was also occupied by lysosomes, mitochondria, and small sheets of smooth ER (sER) that contacted the LD scaffold (Fig. 4B). 3D reconstruction of liver tomograms illustrated the close relationship of sER, LD, and glycogen; here, the sER was positioned within 1-2 nm from glycogen and LDs, sometimes "sandwiching" glycogen between two membranes (Fig. 4C, D). We validated these results by data mining a previously published hepatocyte 3D EM dataset[12], in which glycogen is clearly seen outlining the periphery of an LD (Supplementary Fig. 6C, D). LD-glycogen interactions were also observed in *C. elegans* intestinal cells (which function as the worm's "liver") and human hepatocytes, suggesting that this functional compartmentalization is conserved across different species (Fig. 4E, F). Glycogen clusters were made of small electron dense spheres of ~40 nm in diameter (consistent with the size of glycogen beta particles[52]) and were defined marked by a soft white halo marking the "shell" space occupied by glycogen in situ before sample dehydration for eTomo imaging (Fig. 4B). This led us to the recently proposed hypothesis that glycogen forms and behaves as liquid condensates that trap signaling molecules[48]. Liquid-phase 'organelles are hypothesized to grow by coalescing into each other, which creates an exponential growth pattern of larger droplets absorbing smaller ones[53]. Indeed, glycogen growth appears to follow this predicted pattern in situ, as revealed by volume reconstruction of $n = 150$ individual glycogen objects (Fig. 4G).

Together, these results establish spatial, structural, and molecular features of glycogenesis to indicate that this process occurs on and around the scaffold of hepatocyte LDs.

## Glucose influx reorganizes hepatocyte organelle contact networks during glycogenesis

Organelle function inside the cell can be heterogeneous and dependent on the nature of organelle-organelle contacts[6]. For example, in hepatocytes, LD-associated mitochondria have distinct protein expression patterns and are more adept for fatty acid oxidation versus other "cytosolic" mitochondria[15], whereas ER-associated mitochondria are important for normal insulin signaling and ApoB/VLDL synthesis and secretion[10,17]. To get a broad overview of underlying hepatocyte organelle-organelle architecture, we determined the centroid position of individual organelles in fasted and glucose-infused hepatocytes to reconstruct a network of connected organelle nodes (Fig. 5A). Hepatocytes had a total of ~400-600 nodes each and node connectivity was significantly higher in fasting versus glucose-infused mice (~50% vs 30%, respectively). Glucose infusion decreased the average organelle network size, reducing node connectivity and increasing the fraction of spatially isolated organelles (Supplementary Fig. 8A–D). Spatial network

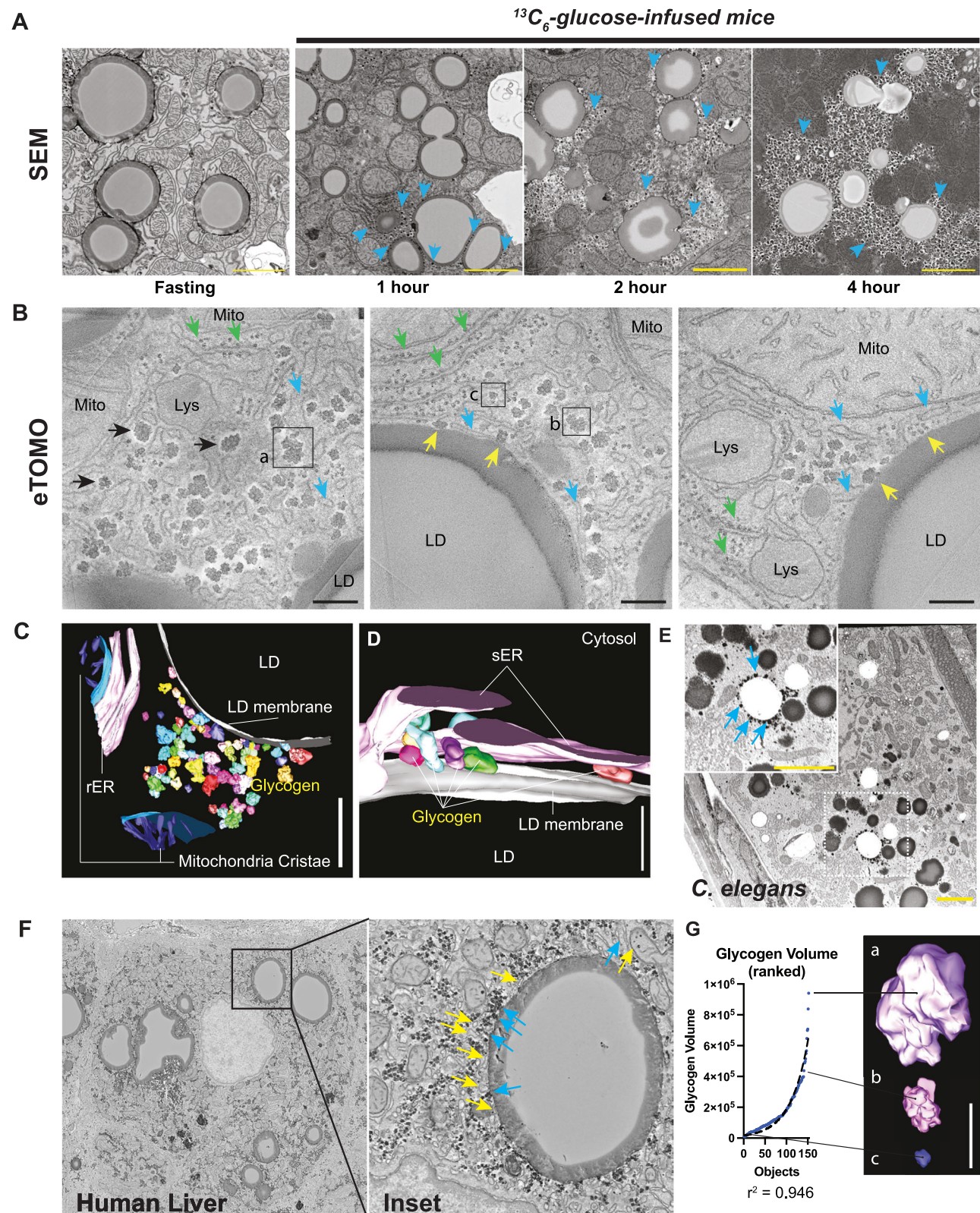

analysis identified organelle first neighbors to reveal the complex landscape of homotypic and heterotypic organelle contacts in hepatocytes (Fig. 5B). In fasted hepatocytes, most organelles are within range of ER nodes, which in turn are in contact with most LD, mitochondria, and glycogen structures. We also observed a fraction of LD-connected glycogen nodes. Moreover, hepatocyte networks from glucose-infused mice were re-organized from ER- towards

glycogen-centric contacts, thus revealing a significant shift in organelle connectivity during glycogenesis (Fig. 5B).

Organelle positioning inside cells is regulated by interactions with the cytoskeleton and is very dynamic across spatiotemporal scales[20,54,55]. To determine if organelle positioning within hepatocytes was stochastic (or not), we applied nearest neighbor analysis and Monte Carlo simulation methods to analyze the overall spatial

**Fig. 4 | Lipid-droplets (LDs) act as scaffolds for glycogen synthesis.**
**A** Representative hepatocyte SEM micrographs showing LDs and clustering of glycogen crystals around LDs in overnight-fasted mice or in mice at 1-, 2- and 4-hour [U-$^{13}$C$_6$]-glucose-infusion timepoints. Blue arrows point to glycogen depots.
**B** Representative electron tomography (eTOMO) micrographs of mouse hepatocytes after 1 hour of [U-$^{13}$C]-glucose infusion, with mitochondria (Mito), lysosomes (Lys), lipid droplets (LD) in view. Yellow, blue, and green arrowheads point to the location of LD-tethered glycogen, smooth endoplasmic reticulum (ER), and rough ER, respectively. **C, D** 3D reconstruction of eTOMO volumes and organelle structures showing the close association between glycogen, ER, and LD. Two partial reconstructions of mitochondria and mitochondria cristae are also shown.

**E, F** Representative scanning electron microscopy (SEM) images of a C. elegans intestinal cell and human male hepatocyte, respectively. SEM insets highlight the position of a single lipid droplet and blue arrowheads indicate the location of LD-associated glycogen particles. In the human micrograph, yellow arrowheads mark the position of smooth endoplasmic reticulum (sER) compartments. **G** Graph displaying reconstructed volume of $n = 150$ glycogen particles imaged with eTOMO and ranked according to their volume (blue dots). Dotted black line indicates the fit of an exponential curve with an observed $r^2 = 0.946$. Insets *a, b,* and *c* show representative glycogen particles of different sizes, and their location are marked in the micrographs shown in (**B**). In (B), scale bar = 50 nanometers, in (**C, D** and **F**), 100 nanometers. In (**A, B** and **E, F**), data representative of $n = 3$-to-5 mice per condition.

distribution of mitochondria, LDs, and glycogen nodes. This analysis determined that glycogen distribution is not random and clustered (likely due to its granular landscape), whereas LDs and mitochondria are randomly distributed within the cell cytosol (Fig. 5C). This data provides a sub-micron resolution map of organelle contact networks and how these are modulated during fasting and the transition towards active glycogenesis.

One of the caveats of our centroid-based analysis is that it fails to properly represent the elongated morphology of organelles (i.e., ER) and therefore it can underestimate the true landscape of organelle contact networks. To address this gap and to identify organelle contact sites at nanometer resolution, we developed a vector-based search algorithm to map neighboring pixels in the SEM and organelle segmentation masks to identify likely areas of organelle contact within 5-to-10 nanometers in distance (Fig. 5D). We identified changes in the size of mitochondria contacts with ER, LD, and glycogen (Fig. 5E–G, and Supplementary Fig. 8E). Given the established role of mitochondria and ER interactions in hepatocyte function, we focused our analysis on these two organelle types and estimated the density of direct organelle contacts and relative changes in organelle contact size and frequency. We analyzed mitochondria-ER contact networks during fasting, randomly fed, or $^{13}$C-glucose-infused conditions at animal, single cell, and organelle scales. Fasting associated with a 1.5-to-7-fold increase in mitochondria-ER contacts in relation to glucose-infused or fed mice (Fig. 5E–I). Notably, 1 hour of glucose infusion led to a ~80% decrease in mitochondria-ER contacts (Fig. 5G) despite small differences in glycogen content and no differences in cell size (Fig. 3K, Supplementary Fig. 8F). This phenotype was sustained after 4 hours of glucose infusion (Fig. 5E–H). These results support that changes in energy demands and nutrient signaling modulate mitochondrial structure-function and organelle contacts[19,35], and indicate that loss of mitochondria-ER contact sites occurs within the first hour of an increase in circulating glucose levels and active glycogenesis.

Next, we quantified the $^{13}$C enrichment levels in ER-connected mitochondria compared to the other types of mitochondria in our dataset (i.e., isolated, or connected to LDs (Mito-LD) or glycogen (Mito-Gly)). This revealed that mitochondria $^{13}$C enrichment was heterogeneous and dependent on the identity of the mitochondria-interacting organelle (Fig. 5J and Supplementary Fig. 8G). Finally, to determine the spatial distribution pattern of different type of mitochondria, we repeated our Monte Carlo analysis and discovered that most mitochondria are randomly distributed within the cytoplasm, except ER-connected mitochondria in fasting hepatocytes (Fig. 5K).

These experiments highlight the use of large-scale image segmentation tools, spatial mapping, and mathematical modeling to investigate patterns of organelle organization and accumulation of glucose-derived $^{13}$C elements in response to changes in nutrient intake. We established that most mitochondria are randomly distributed within the subcellular space, and that ER-contacting mitochondria are non-randomly organized - thus implying that metabolic processes activated (or suppressed) during fasting likely regulate the organization of hepatocyte mitochondria-ER connections.

## Limitations of this study
In this study, we introduce a multi-modal analysis pipeline to quantify nutrient metabolism and channeling across the mesoscale, from whole animals to single cells and organelles. This is achieved by combining in vivo measurements of glucose oxidation and metabolism with MALDI-MS and MIMS-EM imaging, followed by spatial analysis tools to map the flux of glucose-derived $^{13}$C into distinct cellular and sub-cellular compartments. While measuring in vivo animal metabolism is relatively affordable and available in multiple institutions, MALDI-MS and MIMS-EM are very expensive and time-consuming techniques, which limits sample throughput. Importantly, we performed MALDI-MS and MIMS-EM on tissues from at least n = 3-to-7 animals infused with different doses of glucose for up to 4 hours and focused on hepatocytes due to their well-established role in glucose storage and homeostasis processes. This allowed us to analyze large tissue sections (~3 mm$^2$ with MALDI-MS) and hundreds of cells and thousands of individual organelles (~300–500 um$^2$ with MIMS-EM) to identify the sub-cellular location of organelles and their interactions. Moreover, due to the physics of stable isotope imaging and detection of MIMS, MIMS-EM is unable to identify the molecular identity of most molecules associated with the spatial patterns of $^{13}$C distribution, except perhaps glycogen and saturated fatty acids that form lipid droplets (i.e., triacylglycerol and cholesterol esters). While current MALDI-MS techniques are limited to 2-to-5 microns in spatial X-Y resolution, our MALDI-MS approach establishes a pipeline to detect stable isotope enrichment in glycogen and glucose metabolites and thus will lead towards future approaches where different classes of molecules (lipids, carbohydrates, proteins) can be measured from the same spatial region. In the future, other orthogonal techniques compatible with SEM imaging (i.e., Immuno-EM and Click-EM[56]) will allow for identification of molecular species associated with channeling of nutrient types and offer higher spatial and temporal resolution to correlate with MALDI-MS and/or MIMS-EM datasets.

## Methods
### Animals
All animal experimentation was approved by the Institutional Animal Care and Use Committee at Vanderbilt University (IACUC protocols M2000086-00 and M1500013-02). 8-to-24-week-old mice (C57/BL6 males from Jackson Labs (JAX), Connecticut, strain number 000664) were used and maintained in rooms with an average temperature of 23 °C and with a 12 h light and 12 h dark cycle.

### Human liver samples
Wedge liver biopsies of the left lateral lobe of the liver were obtained at the time of elective bariatric surgery. Subjects gave informed written consent before participating in this study, which was approved by the Internal Review Board of Vanderbilt University (171845) and registered at ClinicalTrials.gov (NCT03407833). Studies were conducted in accordance with NIH and institutional guidelines for human subject research. The study protocol conformed to the ethical guidelines of the 1975

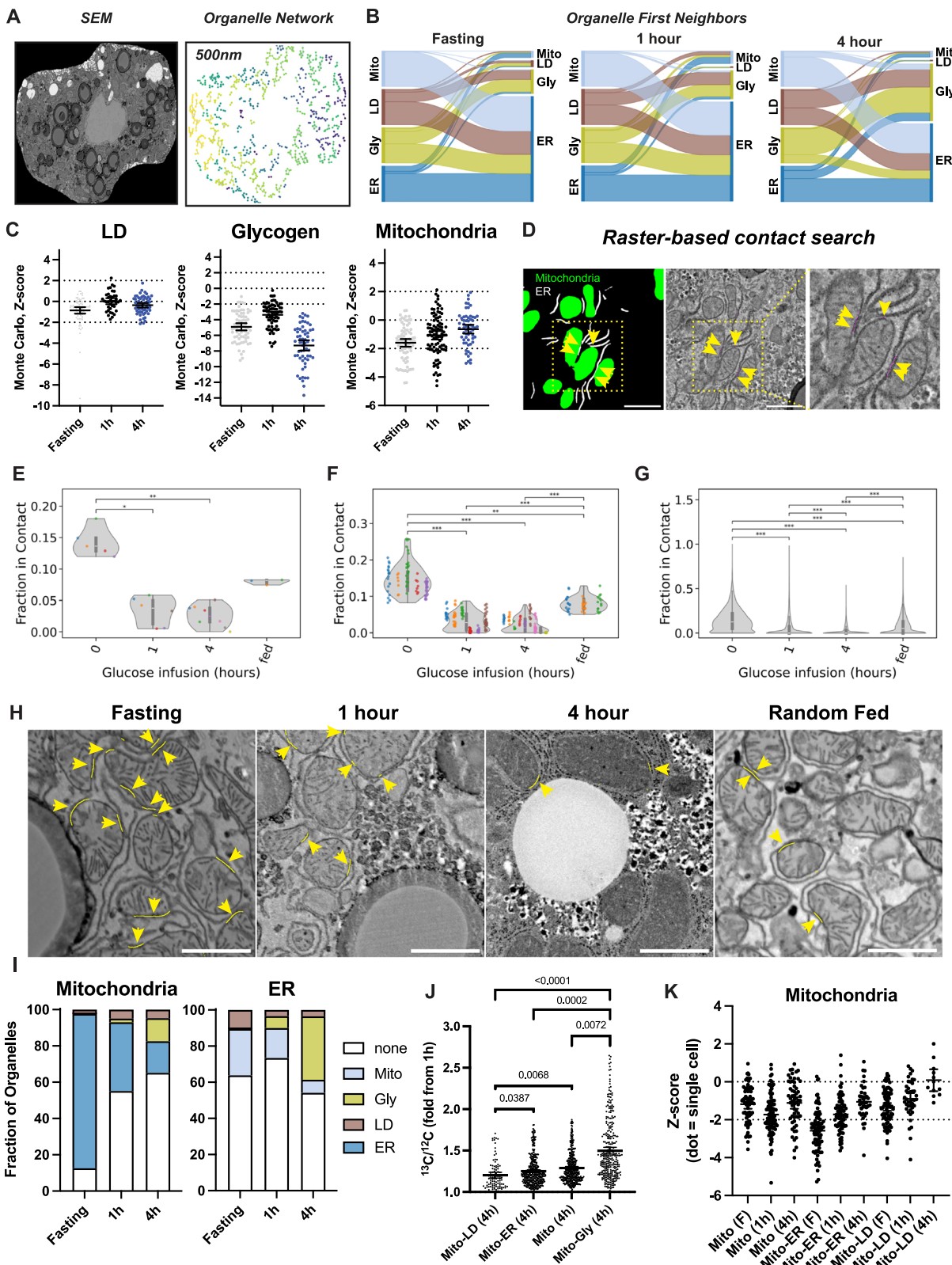

## Catheter implantation and glucose infusion

Catheters were surgically placed in the carotid artery and jugular vein for blood sampling and stable isotope infusions, respectively, as previously established[57]. Catheters were implanted 3 days prior to the

Declaration of Helsinki, as reflected in a priori approval by Vanderbilt University Medical Center.

stable isotope infusions. On the day of the infusions, mice were fasted for 6 h or 16 h before any of the procedures. Mice were awake and were not restrained during the stable isotope labeling experiments. At the end of the infusion, mice were anesthetized with an intra-venous (i.v.) injection of pentobarbital in the jugular line, and tissues were immediately excised and flash frozen in liquid nitrogen for mass spectrometry or prepared for MIMS-EM as described below.

**Fig. 5 | Spatial analysis of hepatocyte or interaction and organization patterns.**
**A** Representative scanning electron microscopy (SEM) and reconstructed organelle interaction network and nodes within 500 nanometers (nm) of each other. Data from a mouse fasted overnight. **B** Sankey graphs displaying the relative proportion of ER, LD, mitochondria (Mito), and glycogen (Gly) nodes that are within 500 nm range of each other in fasting or after continuous infusion with 40 mg/min/kg of [U-$^{13}$C]-glucose for 1 or 4 hours. **C** Scatter plots showing the results of Monte Carlo simulation analysis to determine randomness of LD, glycogen, and mitochondria nodes in each reconstructed hepatocyte spatial network. Each dot represents a cell. Lines at 2 and -2 mark the range where organelle positioning is determined to be random. Dots below -2 indicate cells where the positioning of organelles is clustered and not random. **D** Close up of mitochondria (green) and ER (white) organelles segmented using trained 2D U-nets applied to SEM data. Inset, SEM data where thin magenta lines annotate the location of Mitochondria-ER contact sites. Yellow arrows point to mitochondria-ER contact sites. **E–G** Relative frequency of ER or Mitochondria organelle-contact types after overnight-fasting, 1 or 4 hours of

[U-$^{13}$C]-glucose infusion, or random fed states at mouse, single cell, or organelle level averages – respectively. **H** Representative SEM panels showing annotated hepatocyte mitochondria-ER contacts in fasting, glucose, or fed conditions shown in **E–G**. Yellow arrowheads and lines mark the position of identified ER-mitochondria contact sites. **I** Relative fraction of mitochondria and ER that are isolated, or connected to LD, ER, glycogen (Gly), or mitochondria (Mito). **J** $^{13}$C/$^{12}$C levels by different types of mitochondria classified by the identity of their closest interacting partner after 4 hours of 40 mg/min/kg [U-$^{13}$C$_6$]-glucose infusion. **K** Scatter plots showing the results of our Monte Carlo simulation analysis to determine randomness of isolated mitochondria (Mito) or mitochondria connected to ER (Mito-ER) or LD (Mito-LD) nodes in each reconstructed hepatocyte spatial network. Each dot represents a cell. All data shown with error bars and 95% confidence interval of the data. In **E–G** and **J**, One-way ANOVA with Dunns or Kruskal-Wallis tests where used, respectively. In **E–G**, ***$p < 0.001$, and data from individual animals are shown in different colors, and each dot = 1 animal (**E**) or = 1 cell (**F**). In **J**, $p$ values are shown in the figure.

## Indirect Calorimetry and $^{13}$C-Breath Testing

On the morning of the experiment, mice were individually placed in circular acrylic containers (135 mm internal diameter and 120 mm internal height) with paper-based bedding in a 12 h light/dark cycle, temperature-controlled dedicated room located in the Vanderbilt Mouse Metabolic Phenotyping Center (MMPC, RRID: SCIR_021939). Energy expenditure measures were obtained by indirect calorimetry (Promethion, Sable Systems, Las Vegas, NV). In short, the gas within each container was sampled through a raised microperforated acrylic false floor that ensures uniform gas sampling. Respiratory gases were measured using an integrated fuel cell $O_2$ analyzer, spectrophotometric $CO_2$ analyzer, capacitive water vapor partial pressure analyzer, and barometric pressure sensor (GA-3, Sable Systems, Las Vegas, NV). This system used two GA-3 analyzers operating in parallel, devoted to one container apiece, to maximize metabolic data resolution. Gas sensors were calibrated monthly with 100% $N_2$ as a zero reference and with a span gas containing known concentrations of $CO_2$. The gain of the $O_2$ channel was adjusted during each new incurrent measurement so that incurrent $O_2$, after correction for water vapor dilution and barometric pressure, yielded a concentration of 20.94% STPD (standard temperature [0 °C] and pressure [1ATM], dry). Promethion utilizes a pull-mode, negative pressure system. As such, two multi-channel mass flow generators continually measured and controlled air flows (FR8, Sable Systems, Las Vegas, NV), ensuring that the excurrent flow rates remained constant at 3000 mL/min. Water vapor was continuously measured, and its dilution effect on $O_2$ and $CO_2$ concentrations was mathematically compensated for in the analysis[58]. The $O_2$ consumption (i.e., $VO_2$) and $CO_2$ production (i.e., $VCO_2$) were measured for each mouse continuously. Incurrent air reference/background values were determined every 5 minutes. A 400–600 ml/min subsample of the gas exiting each metabolic chamber was diverted into a stable isotope analyzer (Sable Systems International), which measured the amount of $^{13}$C (i.e., $\delta^{13}$C) in the carbon dioxide in real time using a process called $^{13}$C-breath testing[59,60]. Because the $CO_2$ from the measured gas stream is a dynamic mixture of both ambient $CO_2$ and the $CO_2$ released as a byproduct of the mouse, we used a previously published mathematical approach to remove the effect of ambient $CO_2$ by accounting for the dynamic concentration of ambient $CO_2$ and its $\delta^{13}$C recorded during periodic incurrent air measurements[61]. The $\delta^{13}$C is reported in terms of $^{13}C_{VPDB}$[62]. Respiratory quotient (RQ) was calculated as the ratio of $VCO_2$ to $VO_2$. Energy expenditure was calculated using the Weir equation: EE (kcal/hr) = 60*(0.003941*$VO_2$(ml/min) +0.001106*$VCO_2$(ml/min))[63]. Data acquisition and processing were coordinated by PromethionLive and MacroInterpreter (Sable Systems) using an analysis script detailing all aspects of data transformation. The script is available on request

from Sable Systems. Body composition was determined by NMR (Bruker Minispec).

## Plasma Metabolite and Glycogen Gas Chromatography Mass Spectrometry (GC-MS) measurements

Liver glycogen mass was assessed using the method of Chan and Exton[64]. Here, the final purified liver lysate (in the form of a supernatant) contains glucose moieties derived from glycogen molecules, and quantifying glycogen content based on the deuterium enrichment method as established by ref. 65,66. Plasma metabolites were isolated from 50 µL of plasma using a biphasic methanol/water/chloroform extraction. Norvaline (20 uL of 5 mM) was added to each sample as an internal standard. The polar layer of the extract was isolated using a fine-tipped pipette and air dried overnight for storage at −80 °C prior to derivatization. Polar metabolites from plasma extracts were converted to their methoxime tert-butylsilyl derivatives (TBDMS) using MtBSTFA+1% TBDMCS (catalog 1-270144-200, Regis Technologies). Calibration standards with known amounts of each metabolite were prepared and derivatized simultaneously with the extracted samples for absolute quantification of metabolite abundances. Derivatized samples were injected onto a HP-5ms column (catalog 19091S-433, Agilent Technologies) in an Agilent 7890B gas chromatograph paired with an Agilent 5977 A mass spectrometer. Data were acquired in scan mode, and $^{13}$C- vs $^{12}$C-labeled metabolites were identified through comparison of mass spectra using a previously generated standard library. The accuracy of our mass isotopomer distribution (MID) measurements was validated through comparison of the theoretical and experimental values of unenriched control samples.

## Tissue preparation for MALDI-IMS

Mouse livers were harvested and embedded in carboxymethylcellulose (CMC), flash frozen in a bath of isopentane and dry ice and stored in a −80 °C freezer[67]. Frozen mouse liver tissues were mounted on cold chucks using CM3050 S cryostat (Leica Biosystems, Wetzlar, Germany). The tissue sections were cut at a thickness of 10 µm using a Leica CM3050S cryostat (Leica Microsystems GmbH, Wetzlar, Germany) and immediately mounted onto a cooled ITO-coated slide Delta Technologies (Loveland, CO). Sections were stored in a vacuum desiccator for a minimum of 1 hour. (Background/ Desalting Removal) Lipids were removed from the tissue sections by immersing them in xylene twice, for 3 minutes each time. The sections were rehydrated through a series of organic washes: one wash in 100% ethanol (1 minute), three washes in Carnoy solution (3 minutes each), one wash in 100% ethanol (1 minute), one wash in 95% ethanol (1 minute), one wash in 70% ethanol (1 minute), followed by two washes in 150 mM ammonium formate for 3 minutes each. After rehydration, the slides were placed in a slide mailer containing citraconic anhydride buffer for antigen retrieval, and the slides

were heated for 30 minutes in a vegetable steamer. The citraconic anhydride buffer was prepared by adding 25 μL citraconic anhydride to 50 mL of water, adjusting the pH to 3.0 with HCl, 12 M. After cooling, the buffer was gradually replaced with water through five buffer exchanges, followed by a complete water replacement during the final exchange. The slides were then dried in a vacuum desiccator prior to enzymatic digestion[68].

## In situ isoamylase enzyme application and digestion

A TM-Sprayer (M3, HTX Technologies, Carrboro, NC, USA) was used to apply 200 μL of an aqueous isoamylase solution (3 units/slide)[37]. The spray nozzle was heated to 45 °C, with a spray velocity set at 900 mm/min. Details of the TM-Sprayer-specific parameters can be found in Supplementary Table 1. Following enzyme application, the slides were incubated at 37 °C for 2 hours in a humidified chamber and subsequently dried in a desiccator before matrix application[68]. Before IMS sample preparation, autofluorescence images were captured using standard DAPI, eGFP, and DSRed fluorescent filters. A Zeiss AxioScan.Z1 slide scanner (Carl Zeiss Microscopy GmbH, Oberkochen, Germany) equipped with a Colibri7 LED light source was used for imaging. To apply the matrix, a solution of α-cyano-4-hydroxycinnamic acid (CHCA) (5 mg/mL in 1:1 acetonitrile/water solution with 0.1% TFA) was applied using the TM-Sprayer (M5). The TM Sprayer specific parameters are listed in Supplementary Table 2. After matrix application, the slides were stored in a vacuum desiccator until data acquisition. Following MALDI IMS data acquisition, a post-IMS autofluorescence image was obtained from the tissue section before matrix removal using a Zeiss AxioScan.Z1 fluorescence slide scanner, employing the previously mentioned eGFP fluorescence filter and a monochromatic brightfield image[69]. After capturing the post-IMS autofluorescence image, the sections were stained using a Hematoxylin and Eosin stain following a standard protocol[69].

## MALDI Imaging Mass Spectrometry (IMS)

MALDI IMS of mouse liver sections was performed using a raster step size of 20 μm on a timsTOF fleX instrument equipped with micro-GRID, utilizing timsControl v6.0.0 and flexImaging v7.5 software (Bruker Daltonics, Billerica, MA, USA). Tissue sections were analyzed in positive ion mode, with a mass-to-charge ratio ($m/z$) range of 500–3,500. The Instrument specific parameters are listed in Supplementary Table 3. Prior to data acquisition, external $m/z$ calibration was performed using red phosphorus in positive ion mode. Data were processed in SCiLS Lab 2024b Pro (version 2020 Pro, SCiLS GmbH, Bremen, Germany) to generate glycogen ion images, normalized to the total ion current (TIC) signal for each pixel, without applying the de-noising function. Ion identities were assigned based on accurate mass using a 5-ppm tolerance. Further identification of selected ions was performed using tandem mass spectrometry to validate accurate mass measurements, with collision energies ranging between 80 and 100 eV.

## Sample processing for MIMS-EM

Tissue processing for MIMS-EM imaging was done as previously described by us[24,25]. First, stable isotope-labeled animals were sedated using sodium diethylbarbiturate (Nembutal, 35 mg/kg i.v.) and euthanized via diaphragm resection followed by transcardiac perfusion 37 °C Ringer's solution (0.79% NaCl/0.038% KCl/0.02% MgCl$_2$·6H$_2$O/0.018% Na$_2$HPO$_4$/0.125% NaHCO$_3$/0.03% CaCl$_2$·2H$_2$O/0.2% dextrose/0.02% xylocaine) for 60 seconds followed by perfusion with ice cold 2.5% glutaraldehyde and 2% PFA in 0.15 M sodium cacodylate for 10 minutes (rate at 5 mL/min). Next, we cut small biopsies samples of the medial liver lobe and prepared them for scanning electron microscopy (SEM) followed by MIMS. Here, the perfusion-fixed liver tissue was cut into ~1mm³ pieces and post-fixed in the same fixative at 4 °C overnight. Next, each sample was washed for 1 hour at room temperature using 0.15 M cacodylate buffer and then post-fixed in 2% osmium tetroxide and 1.5% potassium ferrocyanide solution made with 0.15 M sodium cacodylate buffer. Samples were then thoroughly washed in double distilled water (ddH$_2$O) and placed in a 0.5% thio-carbohydrazide solution for 30 minutes followed by five washes in ddH$_2$O at room temperature. Next, tissue samples were placed in a 2% aqueous osmium tetroxide solution for 1 hour, then extensively washed in ddH$_2$O, and next placed in a 2% aqueous uranyl acetate solution at 4 °C overnight. Samples were again washed with ddH$_2$O and placed into Walton's lead aspartate solution and baked for 30 min at 60 °C using a bench-top baking oven. Baked samples were washed with ddH$_2$O followed by serial dehydration using ice-cold ethanol at 70%, 90%, and 100% EtOH followed by dry acetone (10 minutes each step on ice). Dehydrated tissues were placed into 1:3, 1:1, and 3:1 solutions of Durcupan ACM:acetone for 12 hours in each concentration for tissue embedding. Embedded tissues were exposed to three changes of 100% Durcupan ACM for 24 hours each before being baked for 48 hours at 65 °C for solidification.

## Correlative Electron Microscopy and Multi-isotope Mass Spectroscopy (MIMS-EM)

MIMS-EM measures spatially localized concentrations of several isotopes in biological samples overlaid with high-resolution scanning electron microscopy (SEM) to provide accurate spatial and quantitative information regarding the chemical composition of macromolecules, organelles, cells, and tissues[24–26]. To create large field-of-view (FoV) maps of distinct liver lobe regions, 80 nm-thick sections were cut using an ultramicrotome (Leica UC7) and arranged on 5 x 7 mm silicon wafers (Electron Microscopy Services (EMS), cat# 71893-10) and mapped using SEM (Crossbeam 550, Zeiss, Germany). User-supervised image acquisition was guided using automated tile acquisition and image mosaicking software (Atlas 5, Fibics, Ottawa, Canada). Images were acquired with a pixel size of 5 nm and covered areas of approximately 300um² per tissue sample. Next, wafers containing the mapped samples were transferred to a MIMS microscope (50 L NanoSIMS, Cameca, France) for acquisition of multi-isotope maps ($^{13}$C, $^{12}$C, $^{32}$S, $^{14}$N, and $^{31}$P) as previously established[24,28] using the following MIMS image acquisition parameters: image size of 512×512 pixels, raster size of 30-to-40um², at least three frames per raster with a 10 min acquisition time per frame using the beam adaptor D1-3 to yield a spatial resolution of ~80 nm in X-Y.

## Electron tomography (eTOMO) of mouse livers

eTOMO was performed as recently described by us[70]. Here, we cut liver tissue from 1 hour glucose infused mice prepared for MIMS-EM to prepare 300 nanometer-thick sections using a Leica ultramicrotome. Sections were placed on a 100-mesh copper grid, and 20-nm colloidal gold particles were deposited on both sides of the grid to serve as post-hoc image registration landmarks. Samples were loaded into a Tecnai High Base Titan (FEI; Hillsboro, OR) electron microscope operated at 300 kV. Grids were irradiated with electrons for 10 min to limit sample thinning that can occur during imaging data collection. Illumination was held to near parallel beam conditions and the beam intensity was kept constant. Dual tilt series were captured using SerialEM software (University of Colorado, Boulder, CO), and series were taken at 0.81 nm/pixel. Imaging data was detected using a Gatan Ultrascan 4 K x 4 K CCD camera. Each dual-axis tilt series consisted of 121 images taken at 1 degree increment over a range of −60 to +60 degrees followed by a 90° rotation followed by 121 images with the same tilt increment. After data collection, images were binned by 2 to improve signal-to-noise ratios. The IMOD package with etomo java wrapper (https://en.wikipedia.org/wiki/IMOD) was used for tilt-series alignment, reconstruction, volume segmentation, and volume data extraction.

## SEM imaging of *C. elegans*

Sample preparation and imaging *C. elegans* followed the protocol from Belanger et al.[71]. *C. elegans* were cultured on OP50-1 E. coli lawns on standard nematode growth media. A pellet of live, day-1 adult *C. elegans* was resuspended in 0.15 M sucrose prepared in M9 buffer and loaded into 200 μm deep well of a A-type carrier (Leica). The assembly was covered with flat side of B-type carrier (Leica) and vitrified using high-pressure freezing machine (Leica EM ICE). The frozen specimens were stored in liquid nitrogen until further processing. The freeze-substitution (FS) process was performed using a cocktail of 1% osmium tetroxide in acetone. Briefly, samples were transferred into automatic freeze-substitution machine 9AFS 2, Leica) and kept for 9 hours at −90 °C, warmed up over period of 12 hours to −20 °C and then again warmed up over period of 5 hours to 0 °C. The samples where then transferred into an ice filled bath and incubated for 20 min in solution 1 (FS cocktail: ultrapure water = 3 :1), 20 min in solution 2 (FS cocktail: ultrapure water = 1 :1) and then 20 min in solution 3 (FS cocktail: ultrapure water = 1:3). Following this, samples were incubated in 1% osmium tetroxide in 0.1 M cacodylate buffer for 1 hour and during the incubation period transferred to a room temperature. Osmium solution was then removed, and samples were incubated with 2.5% potassium ferrocyanide in 0.1 M cacodylate buffer for 1 hour. Samples were washed 3 times for 15 min each with ultrapure water and incubated in 1% thiocarbohydrazide solution at 60 °C for 30 min. Following this, samples were washed 3 times for 15 min each with ultrapure water and incubated with 2% osmium tetroxide for 1 hour at room temperature. Samples were then washed again in ultrapure water and incubated with 1% aqueous uranyl acetate at 4 °C overnight. The next day, samples were moved to 50 °C for 1.5 hours. After that samples were washed in ultrapure water 5 times for 5 min each and incubated with 20 mM lead aspartate for 1 hour at 50 °C. Samples were then washed 3 time for 10 min each with ultrapure water, dehydrated in a graded acetone series (50%, 70%, 90%, 100% x3) for 10 minutes in each step, and infiltrated with microwave assistance (Pelco BioWave Pro, Redding, CA) into Durcupan resin (Electron Microscopy Sciences). During final steps of resin infiltration, individual worms were transferred from carriers into resin molds with a fine needle, and samples were cured in an oven at 60˚C for 72 hours. Post resin curing, specimens were exposed with a razor blade, and 70 nm thin sections were prepared on silicon wafer chips. These chips were then adhered to SEM pins with carbon adhesive tabs and specimen cross-sections were imaged in a FE-SEM (Zeiss Merlin, Oberkochen, Germany) at 8 kV and 900 pA to identify regions of interest for further analysis. Once regions of interest were identified, resin blocks were mounted onto SEM pins with silver epoxy and sputter-coated with 6 nm of iridium (Leica ACE 600, Vienna, Austria). Samples were then loaded into Helios 5 UX DualBeam (Thermo Fisher Scientific, Brno, Czech Republic) and regions of interest were located by secondary electron imaging at 5 kV and 800 pA. Serial block-face imaging was performed at 1.8 kV and 800 pA using the ICD and TLD detector in backscattered mode and the ASV 4 software (Thermo Fisher Scientific, Hillsboro, Oregon, USA). The block was milled at a current of 750 pA with 10 nm slices, and images were acquired at a resolution of 10 nm/pixel with a dwell of 6 μs and a line average of 2. A stack of acquired images was aligned using Amira 2019.4 (Thermo Fisher Scientific).

## MIMS-EM Image Registration and Data Display

The elemental maps and relative isotope ratios (i.e., $^{13}C/^{12}C$ ratiometric images) obtained by the 50 L NanoSIMS were overlaid on the SEM image of the same section, after alignment and post-processing to create an overlay image that contains both the elemental maps from MIMS and the spatial resolution of the SEM. MIMS-EM imaging data registration was performed using a Python-based version of the "MesoFusion" plugin tool in ImageJ[25]. Briefly, each MIMS image was re-scaled to match the pixel size of the corresponding SEM image, and

coarsely aligned using linear image transformations (linear shifts, orientation, and image flips) to ensure both MIMS and SEM images have similar fields of view. Next, we applied either manual or machine learning (ML)-based image segmentation to annotate matching landmark structures on both SEM and $^{32}S$ MIMS images to create fiducial points that were used to guide image registration using UnwarpJ[72]. Next, we applied the transformation matrices to the $^{13}C/^{12}C$ ratiometric images and binned the $^{13}C/^{12}C$ ratiometric data into categories representing different levels of relative $^{13}C$ enrichment to create MIMS-EM overlays. The CMYK coloring pattern was generally followed to promote a color-blind friendly visualization of all imaging data. Quantification of $^{13}C/^{12}C$ ratiometric signal at the cell level was calculated by manually drawing around individual cells or, for organelles and sub-cellular neighborhoods, data was automatically extracted using spatial analysis software (described below).

## Machine learning (ML) segmentation of organelles

To create segmentation masks of hepatocyte endoplasmic reticulum (ER), glycogen, mitochondria, and lipid droplets (LDs) imaged with SEM, we trained 2-dimensional (2D) U-nets using Aivia software's Deep Learning analysis module (Leica Microsystems). 2D U-nets were trained using multiple sets of manually annotated SEM images. Each training set consisted of an 8-bit SEM image and a matching 8-bit binary mask image of a specific type of organelle, which was called the ground truth (GT). GT images were created using manual annotation of features (e.g., ER sheets, mitochondria) using the LabKit plugin on ImageJ. At least 25 different pairs SEM and GT images were loaded into Aivia (v11.0), and 2D U-nets were trained with the following general hyperparameters: 8 layers, 64 Init Filters, 64 Filter Growth Factor, a channel reduction factor of 8, an image block size of 256×256 pixels, and an intensity threshold of 0.25 and area ratio threshold of 0.05 for foreground path selection. We used the Adam optimizer with a learning rate of 0.0001 and a staircase exponential decay for the learning rate scheduling method. The number of epochs for each model ranged from 600 (Mitochondria, LD, and glycogen) to 10,000 (ER) with 256 steps per epoch. Balanced Binary Cross Entropy was used as the loss function. The resulting trained models were applied in batch to SEM images of single hepatocytes manually segmented using ImageJ to create 32-bit organelle "probability maps". After optimization to determine the ideal probability interval for each organelle, each probability map was thresholded before converting each image to 8-bit binary masks for further processing. All imaging data was processed using one of two standalone computers configured with a 14-core Intel Xeon W-2275 with a 3.3 GHz CPU, 256GB of DDR5 RAM, and GPU cards with 24 or 48 GB of memory (RTX-Quattro 5000 or 8000 series, respectively).

## Optimization and benchmarking of ML segmentation models

The accuracy and precision of each segmentation model were optimized and evaluated using two different approaches. First, we determined the optimal threshold level for our organelle segmentation model by plotting a line profile across representative organelle planes and determined that a confidence interval of 90% was sufficient to isolate most true-positive pixels (Supplementary Fig. 4A). Second, we applied our image segmentation models to the SEM images used to train each model and create probability maps which were then thresholded at different levels (50, 70, 80, 90, and 99%) and compared to the manually annotated GT masks. Comparison of ML segmentation and GT image pairs was done using CellProfiler (version 4.2.6) the "*MeasureObjectOverlap*" function to extract f-scores, precision, *recall*, and the "*MeasureColocalization*" function to quantify image colocalization indexes (Pearson's and Mander's) for each ML segmentation model. A confidence interval threshold cut-off of 90% was used to create segmentation binary models since this confidence range achieved a high level of precision and object colocalization (-60-to-

90%) by successfully isolating true-positive pixels (object recall rates of ~70-to-90%) while sustaining a small (~5%) *false-positive rate* (Supplementary Fig. 4A-B). A representative set of manually annotated images with a sample size of 58-to-98 image pairs was used for benchmarking our ML models.

## Pre-processing of MIMS-EM for spatial analysis and quantification

We used M**ATLAB** (Version 2020b, MathWorks) to develop a computational pipeline to automatically process and quantify several spatial aspects of cell- and organelle-anatomy and $^{13}$C enrichment levels acquired with SEM and MIMS, respectively. Since our models have a ~5% false-positive rate (Supplementary Fig. 4A, B), we implemented steps to eliminate any eventual pixel overlap across organelle segmentation masks and improve segmentation accuracy. First, any holes inside objects (i.e., cell and mitochondria masks) using MATLAB's "*bwfill*" function and "*holes*" argument. Next, to smooth organelle shapes, an erosion filter was applied 4x to ER masks using a 3×3 "square" morphological structuring element and MATLAB's "*imerode*" function. Furthermore, using MATLAB's "*imdilate*" function and a 3×3 "diamond" structuring element, we applied 2x a dilation filter on the LD masks. Next, we take a negative of the cell mask, containing all the pixels outside of the cell, and store it in a multi-dimensional matrix as the first image to ensure all image objects analyzed are inside the cell area. The other masks are stowed in the following order: nucleus, mitochondria, ER, Glycogen, and LD. Next, the $i^{th}$ image in the order is subtracted by the 1$^{st}$ through the ($i^{th}$ − 1) image to ensure no overlap between any of the organelles. This organelle organization and subtraction order were decided by visually comparing the end results of the subtractions to the original EM images as a measure of segmentation improvement. After the first cycle is complete, using the subtracted images, the cytosol mask is subtracted first by the nucleus, followed by mitochondria and LD masks. To compensate for the spatial resolution difference between SEM and MIMS images and enhance MIMS-EM signal-to-noise, we applied a mean filter of 17 pixels (80 nm in size) to the $^{13}$C/$^{12}$C images. Rare $^{13}$C/$^{12}$C image pixels values that were below 102 were reset to the terrestrial background value of 102. Next, once all organelle masks were pre-processed, we identified the spatial position of individual organelles using MATLAB's "*bwconncomp*" function that uses a connected component labeling algorithm for binary images. To filter remaining false-positive objects, we applied an object size exclusion filter using pre-determined threshold sizes for each organelle class object (ER, 500 pixels; Mitochondria, 5000 pixels; Glycogen, 500 pixels; LD, 5000 pixels). The size of these exclusion filters was determined after calculating the size distribution associated with each unique object and organelle class. Finally, after all image and mask preprocessing was achieved, the filtered and processed organelle image masks were converted to arrays containing pixel-by-pixel metrics that reported the spatial location of all remaining pixel values and their respective $^{13}$C/$^{12}$C ratiometric values.

## Spatial Analysis of MIMS-EM data – raster scan algorithm

We extracted morphological features (i.e., circularity, area, perimeter, total size, X and Y coordinates of object centroid) from all segmented organelles. Organelle processing was achieved by mapping the centroid of individual organelles and was stored in indexed matrices along with organelle $^{13}$C/$^{12}$C ratios. Individual objects were mapped using 8-pixel connectivity patterns, and a unique object identifier was stored using a 16-bit image format. In addition, we created corresponding matrices to store the spatial coordinates and $^{13}$C/$^{12}$C values for each individual pixel linked to each unique organelle object. Next, we

classified and mapped organelle types (i.e., mitochondria, endoplasmic reticulum, glycogen, and lipid droplets) to individual objects according to the identity of the segmented image. We calculated organelle distances using the Euclidean distance formula:

$$d(p,q) = \sqrt{\sum_{i=1}^{n}(q_i - p_i)^2} \qquad (1)$$

Here, the distance $d$ between points $p$ and $q$ in $n$-dimensional Euclidean space can be defined in terms of $q_i$ and $p_i$, or the cartesian coordinate component in the $i^{th}$ dimension of the n-dimensional space. To identify "contact sites" between neighboring organelles, we first defined a range of distances as our search radius. In our case, we considered pixels to be "in contact" with one another between different organelle objects if the distance between two pixels was within 1–2-pixel lengths, or 5-10 nm. Next, we centered on the pixels lying along the perimeter of each organelle object and calculated relative distances between neighboring organelles. To reduce the time needed for computations, distance calculations were only performed on a "region of interest" (in 2D, a [m + 1] x [m + 1] matrix where "m" is the maximum of our "search threshold" in pixels) centered on said perimeter pixel. If a neighboring organelle image object contained pixels within the search distance radius, then these pixels were annotated as contacting pixels (see attached code for more details). To quantify the total area occupied by organelle contacts in each organelle, features such as the number of "contacting" pixels and the perimeter of an object in contact, for example, are recorded. To denote separate contact sites from one another, MATLAB's connected component labeling algorithm, '*bwconncomp*' is used in a similar fashion to how organelle objects are labeled previously. To reduce the false positive rate, contact sites of a single contacting pixel were removed. Organelle perimeters were obtained using 2D image convolution with MATLAB's "*conv2*" function and a kernel to detect 8-connected image edges. Perimeter length is reported as the number of pixels that lay along the perimeter, rather than the pixel length distance, thus, quantification of the percentage of an organelle objects perimeter in contact with another is reported as the ratio of the number of contacting pixels to the number of pixels along object's perimeter. To quantify the composition of each cell by organelle area, following the processing of the 8-bit organelle masks, the remaining area covered by each organelle type is compared to the area covered by the manually segmented cell mask. To obtain the average $^{13}$C/$^{12}$C value for organelle image objects and per organelle type across an image, we used the arithmetic mean of the $^{13}$C/$^{12}$C image pixel values in the corresponding spatial regions of interest between the different images. When calculating the distances between image objects, the centroids of the image objects are used. Circularity scores were calculated according to the formula:

$$\text{Circularity Score} = \frac{(4\pi) \cdot A}{P^2}(1 - 0.5r)^2 \qquad (2)$$

Here, A is the area of the object, P is the object perimeter, and:

$$r = \frac{P}{2\pi} + 0.5 \qquad (3)$$

## Spatial Analysis of MIMS-EM data – cell node networks

Cell networks were reconstructed using Python 3.6 using the following packages: *numpy* and *pandas* were used to import data, *network* was used for network analysis and node connections, and *matplotlib* was used for creating output plots. Briefly, we generated a comma separated value (CSV) output from our raster scan analysis for each cell that contained the X and Y spatial coordinates and organelle classification according to their class (e.g., Mitochondria, LD) or organelle classified according to their contact type (e.g., Mito-LD, ER-LD). For all spatial

analysis, we removed X-Y points representing the contact sites and the nuclear envelope. Next, connected node networks were built using the networkx's *cKDTree* algorithm to establish node connections within a set distance of 500 nanometers, and connected nodes that were part of the same network were annotated as being part of a unique "community". Next, unique communities were classified according to the number of connected nodes (e.g., 1to3, 4to10, 10+). Nodes not connected were classified as isolated. Finally, we ran a summary pipeline to collect overall information on organelle distribution within the generated cell networks and relative organelle composition per cell. Next, we performed a nearest neighbor analysis where each nearest-neighbor distances were calculated by building a KD-tree using the *scipy.spatial.KDTree* algorithm to map nearest-neighbor distance relationships for each organelle node in the network. Here, we determined the average nearest-neighbor distance per organelle node and compared that index to an expected mean distance for a completely spatially random (CSR) distribution using Monte Carlo simulations. The expected mean was calculated based on a point density within the defined spatial area. Next, the ratio $R = (observed\ mean\ distance\ / expected\ mean\ distance$ was used to determine whether objects were clustered ($R < 1$) or dispersed ($R > 1$). For the Monte Carlo test, we used 1000 simulations per point using random points generated within an area defined by a bounding box within all spatial points using the formula $area = (max_x - min_x) \times (max_y - min_y)$, followed by nearest-neighbor distance calculations for each generated point. We then calculated the mean and standard deviation of each simulated nearest-neighbor distance and calculated a *z-score* to quantify the data deviation regarding each node clustering from stochastic distribution. Finally, the z-scores for each cell and organelle type were plotted as scatter plots.

### Reconstruction of protein-protein interaction (PPI) networks

To reconstruct PPI networks for LD, glycogen, and mitochondria-associated proteomes, we downloaded data from previously published datasets. Protein IDs were used to query the STRING DB and build PPIs with confidence intervals of at least 0.7 (high confidence). PPI networks were then loaded into Cytoscape where the data was analyzed using the functional enrichment clustering function in the STRING DB app. Finally, we filtered the significantly the identified and enriched clusters for the "STRING clusters" metadata and plotted the network graphs using the "*Edge-weighted Spring Embedded Layout*" using the *score* parameter. Nodes that were either isolated or sub-networks that contained less than 3 nodes and were not linked to the main network were removed.

### Statistics

Student's t-test (Prism 10, GraphPad) was used to compare two groups, and One-Way ANOVA was used to compare three or more groups. Here, experimental datasets that failed to display a Gaussian distribution were analyzed using a non-parametric One-way Anova followed by Dunn's multiple comparison test. A p-value of <0.05 was considered significant. In all figures, data is shown with ± 95% confidence interval (C.I.) of the data.

### Reporting summary

Further information on research design is available in the Nature Portfolio Reporting Summary linked to this article.

## Data availability

All source data is provided with this paper. Tomography micrographs, imaging mass spectrometry data and glycogen imaging protocols, and machine learning models can be freely accessed in Mendeley Data (https://doi.org/10.17632/tr8xdwv28g.1). Source data are provided with this paper.

## Code availability

Code pipelines can be freely accessed on GitHub (https://github.com/ArrojoDrigoLab/MIMS-EM).

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

## Acknowledgements

This research was supported by recruitment funds from the Vanderbilt's Department of Molecular Physiology and Biophysics and NIH grants 1R03DK127484, 5U24DK097771, and 1R01DK138141 (RAeD), and R56AG082758 (RAeD and KB), an AFAR/Glenn foundation to KB, and an NIH-NINDS grant U24NS120055 to M.H.E. We are thankful to the team at the Vanderbilt Mouse Metabolic Phenotyping Center (Supported by DK135073 and 1S10RR028101-01), to Dr Evan Kristoflyak for assistance with MIMS-EM sample preparation and SEM imaging performed at the Vanderbilt Cell Imaging Shared Resource (supported by NIH grants 1S10OD028704-01A1, CA68485, DK20593, DK58404, DK59637, and EY08126), to the staff at the Washington University Center for Cellular Imaging (WUCCI) for sample preparation and SEM imaging of *C. elegans* specimens, and to Dr Yunbin Guan at the Division of Geological and Planetary Sciences of Caltech for multi-isotope mass spectrometry (MIMS) data collection.

## Author contributions

A.H., C.A., K-Y.K., A. Z., M. D., S. P., M. C., E. P., K.B., C.R.F., L.L., T.D., O.P.M., J.S., M.H.E., and R. AeD. collected and analyzed data. O.P.M., M.H.E., and R. AeD designed the study, and A. H., C.A., O.P.M., M.H.E. and R. AeD wrote the article. R. AeD is the guarantor of this work.

## Competing interests

The authors declare no competing interests.
