## [Transparent Peer Review file · Nature Communications]

Spatial patterns of hepatocyte glucose flux revealed by stable isotope tracing and multi-scale microscopy

Corresponding Author: Professor Rafael Arrojo e Drigo

Version 0:

Reviewer comments:

Reviewer #1

(Remarks to the Author)

Habashy and colleagues present a new quantitative multi-modality imaging approach involving stable isotope tracers, SEM, NanoSIMS, and sophisticated data processing to study glucose metabolism at high spatial resolution in rodent liver in vivo. This manuscript is ostensibly a methods paper (Figures 1-3 and much of Figure 4) with some potentially interesting biological observations near the end of the manuscript related to organelle contact sites and glucose to biomass flux into organelles parsed as a function of contact site frequency. Given these two aspects of this manuscript, I will address each independently.

From the standpoint of the methodology, the execution of the imaging and data analysis is exceptional, which is not surprising given prior work by the laboratories involved in the project. Many of the elements of the methodology they advance have been demonstrated in prior publications by their groups or other groups, including examples of detecting flux of labeled glucose into glycogen, amongst other subcellular structures, with NanoSIMS. However, they have put these pieces together excellently, including sequential imaging and meticulous image registration (which is no small task). Moreover, the computational work is arguably the most innovative part of the manuscript, particularly when viewed together with the other aspects.

1) A minor point is that the foundations for this work are somewhat under-referenced in my opinion. The correlative SEM and NanoSIMS is an important advance particularly at the resolution and with organellar discrimination achieved by the Ellisman lab. However, there are important examples of EM with NanoSIMS dating back to the early 2000s, and examples of similar approaches used to follow labeled glucose into intracellular biomass including glycogen (e.g. PMID 25262580). The authors have also previously published similar correlative analyses. I think the current accomplishments are not diminished by a more nuanced presentation of prior similar efforts.

2) It seems like cellular data from different animals are merged. This may be reasonable if labeling is similar and normal (although they are not all truly independent). Bimodal distributions are cited (e.g. Figures 4 and S5), particularly at the early timepoint. In this instance, it would be particularly important to ensure that these distributions are bimodal within animals/cells and not just a reflection of animal specific differences in organelle morphometry or labeling efficiency. In addition, it would be helpful to visualize key data such that the different mice can be distinguished.

3) ANOVA and t tests are used, according to the methods, but many of the distributions do not appear to be normally distributed, and so consideration of non parametric tests should be employed when appropriate.

From the standpoint of experimental design and biological conclusions, there are potentially interesting observations contained in Figure 4GH, but they are preliminary and speculative and therefore it is difficult to isolate a specific hypothesis tested or discovery that has been achieved with this exciting methodology (without substantial additional experimentation).

1) There is insufficient consideration of fed controls in my opinion. Why did the investigators not have non-fasted controls for comparison to try to model steady state conditions?

2) With fasting (particularly prolonged fast of 16 hours) followed by reversal of negative energy balance, there are undoubtedly dramatic changes in organelle dynamics in the liver. It is perhaps not surprising that there would be changes in

organelle contact points and label flux, simply due to the shifting biomass and stochastic effects – and if it is stochastic as opposed to a regulated phenomenon, is it still of high biological significance?

3) Most of the comparisons that are statistically significant in Figure 4 are between 1 hour and 4 hours of labeling where significant augmentation of labeling is expected. It seems like the real differences that need to be flushed out are those between different organelle subtypes at the same timepoint. Such differences are referenced, but it was not clear from my reading of the figures/legends if those statistical analyses were performed and what the results were?

4) Differences in progressive label uptake over time, especially in a system that is not at steady state does not necessarily implicate differential turnover rates. Without formally measuring turnover, the discussion of turnover should be reconsidered.

(Remarks on code availability)

Reviewer #2

(Remarks to the Author)

The authors employed MIMS-EM to map the ¹³C distribution at subcellular level in hepatocytes of mice infused with ¹³C-U-glucose. They provided clear evidence that glucose-derived carbon can be mapped to specific organelles. When combined with machine-learning image segmentation, their approach revealed dynamic changes in spatial pattern of glucose ¹³C flux in hepatocytes. They also detected dynamic changes in inter-organelle contacts during the glucose infusion time. A main reservation about the study is insufficient new biology uncovered by the technique. Contrasting with a diabetes model would likely uncover biological insignificance of the morphological observations and increase the impact of the findings. The inability to detect specific metabolites also reduces the significance. Other comments are as follows.

- 1) The proximity of glycogenesis with lipid droplets was shown for brown adipocytes. Is that true also in hepatocytes? What is the biochemical basis for such proximity?
- 2) Figure 1D-E, which infusion rate do these panels correspond to? Please clarify infusion condition for each figure and whether similar findings are made with the different infusion rates.
- 3) Figure legends are missing.

(Remarks on code availability)

Reviewer #3

(Remarks to the Author)

The technique looks promising and relevant, but the authors should also comment on the "throughput-ness" of the technique. The claims of the author sound far-fetched and need to be toned down to reflect the reality of the advance. Generally, the writing style needs to be improved. There are many instances of grammatically and scientifically incorrect sentences that need to be rephrased. This detracts from the paper.

Specific points:

1. What was the rationale for the pentobarbital-based method of anesthesia for tissue collection, especially considering the varied effects of anesthesia and/or euthanasia on rodent tissue metabolism? Refer to following literature: <https://www.ncbi.nlm.nih.gov/pmc/articles/PMC4319778/>
2. Where are the scale bars in Figure 2A-E? The authors must have forgotten,
3. Lines 107-110, "To validate....Figure S2A)": It is not clear what was used for GC-MS, was it hepatic tissue extract? If so, please explain the metabolite extraction protocol in methods section.
4. Figure ...: Define AFE in Figure legend (how is ppm related to AFE?)
5. Line 131-133, "¹³C-labeled glycogen stores....engulfed by glycogen over time...": Statement not clear, what are the authors observing here: glycogen engulfed by what? The sentence refers to Figures 2A and 2G but it is not clear from the images too what is being looked at in terms of time, especially because LDs are not defined (via ROIs/staining) in these images 2A and 2G? Finally how do the authors know they are looking at glycogen.
6. Lines 133-134, "Similar...(Figure 2H-I)": I am not sure what the authors are trying to convince us about? Physical vicinity of glycogen to LD's? Again, where is the LD in Figure 2H? This is a speculative statement at best, and the authors have essentially no proof about enzyme locations. The authors are overreaching.
7. Lines 143-144, "This suggested...and interactomes": This sentence is claimed to be a hypothesis; however, this sentence is the re-statement of a very basic concept that- Glucose metabolism participates in multiple pathways other than glycogen synthesis including amino acid synthesis and fatty acid synthesis. Please change the word "hypothesis" in Line 145. Refer to: Han, HS., Kang, G., Kim, J. et al. Regulation of glucose metabolism from a liver-centric perspective. *Exp Mol Med* 48, e218 (2016). <https://doi.org/10.1038/emm.2015.122>
8. Line 152-153: What is the meaning of "morphological levels in hepatocytes"? The corresponding Figure 4C that has been referred to talks about fraction of cell area. Are the authors trying to say, "relative size of the organelles in the hepatocytes"?
9. Line 153-156: "This revealed....(Figure 4C)": very long and confusing sentence. What is the meaning of "changes to the overall hepatocyte organelle composition"? The statement is grammatically extremely hard to follow and conceptually

unjustified. What about suppression of de novo lipogenesis (triglycerides) in the liver being down-regulated and the effect on lipolysis being unchanged? This statement of lipolysis must be revised and further justified if the authors wish to make it.

10. "Mesoscale channeling" claimed by the authors in the title can only be confirmed by a time-lapse imaging (not just two time points like 1 and 4 h) to see a particular labeled (or MS identified) metabolite getting channeled from one organelle to another. The existence of organelle contacts does not guarantee travel/metabolite transfer. ¹³C enrichment can be due to multiple factors, not necessarily from one metabolite being channeled across organelles.

11. How are the ground truths generated in the SEM images? Is this purely on morphology? If so, the authors should truly show an additional method – fluorescence – to show the specificity of their annotations and DL segmentation.

12. A final comment is in the limitations section. If the method is limited to measuring ¹³C/¹²C ratios, what is the point of the MIMS and mass spectrometry in general? What is the advantage over Raman or vibrational microscopy, which has virtually zero sample prep that even has more chemical specificity and is isotope specific: see

<https://onlinelibrary.wiley.com/doi/full/10.1002/jcb.30372> and <https://www.ncbi.nlm.nih.gov/pmc/articles/PMC6599680/>? The authors must address in their limitations section.

Minor comments:

1. Line 156: "This increase in glycogen" can be replaced by "This increase in the size of glycogen stores".
2. Line 157: Rephrase "loss in LDs...VLDL particles." This part does not make much sense.
3. Line 167, "Within 4 hours...lost to increase....(Figure 4E)": Sentence is unclear, rephrase.

(Remarks on code availability)

No code was made available.

Reviewer #4

(Remarks to the Author)

(Remarks on code availability)

Version 1:

Reviewer comments:

Reviewer #1

(Remarks to the Author)

I have reviewed this new manuscript. The authors have clearly done substantial work in the interim since the index submission and overall have made an effort to respond to the initial reviews. The manuscript is undoubtedly improved, although not having the original manuscript with which to compare (and given the absence of clear designation of which textual components and figure components are new) it was somewhat difficult to fully assess.

My opinion of the quality of the correlative imaging has not changed. It is spectacular.

My lingering concerns about this manuscript are two-fold:

1) The demonstration of inter-organelle contact points may be interesting, but in the current form is underdeveloped. In short, it is not clear that the methodology in this manuscript has been leveraged to make an impactful biological discovery. They may have! -- but it still seems immature from that perspective. For example, if I understand their Monte Carlo analyses, it shows that many of these interactions probably are stochastic, although this does not apply to glycogen. However, does this just suggest clustering of glycogen granules, which is already known? It seems like many of these interactions might be expected based on the dramatic contraction and then expansion of glycogen stores that comes with prolonged fasting followed by a glucose load, not to mention the additional potential confounding factor of intracellular trafficking of newly synthesized glycogen and therefore in its current form it is difficult to know what this really means. I remain intrigued by how the patterns would differ in a fed control state.

2) I continue to have some concerns about the data analyses and merger of cells from different animals. Simply showing that one animal in a relatively small experiment (n=5) is contributing more to the variance than the other 4 animals in my view is not sufficient to exclude it. One could do a formal 'outlier' test, but excluding on that basis in this context would be somewhat controversial. Beyond the question of excluding that animal, cells from different animals should not simply be merged in this way because the data points cannot all be viewed as independent--it seems to be a merger of technical replicates (hepatocytes) from different biological replicates (n=4-5 mice). At the very least, it would seem that there should be a greater effort to reveal the variance across animals in these experiments. They get close with the color coded dot plot shown in the response document, but no such formal analyses or data visualization seems to be present in the manuscript itself.

Regarding Reviewer 2's concerns:

A main reservation about the study is insufficient new biology uncovered by the technique. This remains a concern. They have shared new data, but have not really made a central mechanistic discovery beyond showing associations of new glycogen with different intracellular compartments including lipid droplets. They have reanalyzed MS data and find additional circumstantial evidence (PPIs) to support potential physical/functional interactions between compartments, but I'm not sure demonstrating interactions between growing glycogen molecules and other cytoplasmic structures in isolation is a critical new discovery in isolation.

Contrasting with a diabetes model would likely uncover biological insignificance of the morphological observations and increase the impact of the findings.

They did not do additional models. Without looking at additional models, it is difficult to know what they would add, as this is not necessarily a request that addresses a narrow hypothesis. I agree with the original reviewer that additional data might have provided context. Whether it was the disease model or some other experimental condition that could have yielded a clearer 'discovery' I don't know, but this particular point raised by the Reviewer speaks to the first point above.

The inability to detect specific metabolites also reduces the significance. They did MALDI to show some molecular specificity. However, the MALDI does not add much, because of its low resolution. I don't know of an obvious and direct way to address this question and bring molecular specificity at the resolution of SEM or NanoSIMS. While I don't view the MALDI data as being an important addition, I also don't fully agree with the original question raised by Reviewer 2 that this is an important issue.

Beyond these larger questions of novelty/impact, the specific points raised by Reviewer 2 seem to have been addressed, including have some data in BAT/hepatocytes.

(Remarks on code availability)

Reviewer #3

(Remarks to the Author)

The authors have worked duly on the manuscript and the clarity of writing has significantly improved. This allows for a better appreciation of the technique/science being described. The addition of eTOMO and Monte Carlo simulation analysis are definitely making the paper more convincing. However, there are some places with errors need to be fixed and some points that need to be clarified. These are listed below:

1. Point 3 of my comments has not been responded to in the response doc (despite the authors saying at point 9 that points 3-9 have been resolved). While it does appear that the tissue type used for GC-MS has been clarified in the results section (lines 123-124), however the methods of metabolite extraction employed for GC-MS samples has not been elucidated as requested in the review.
2. Also, line 125 claims Figure S2A to describe the GC-MS data which is not the case. Such errors of mis-referencing of figures are present throughout the manuscript, having a negative impact on the reader and providing a poor impression of the authors and, frankly, the journal.
3. Point no. 4 in the list of comments has not been addressed at all, neither in the response document, nor in the manuscript. Please address
4. Response to point 5 is incomplete. What is the rationale for using the word "engulf"? Is saying that "LD's were profusely surrounded by ¹³C glycogen stores" a better way of phrasing it?
5. Point no. 8 in the list of comments have not been addressed in the response file. It appears from the manuscript file that the figure as a whole and the corresponding text have been replaced but it would be helpful to get a direct point-by-point response from the authors.
6. One of the authors in the original paper "Tom Deerinck" has not been named in the revised manuscript. Has their contribution been removed from the paper or is this a typo?

(Remarks on code availability)

I have briefly reviewed the code - actually the ReadMe.m file, which appears to give instructions on how to use the code.

Reviewer #4

(Remarks to the Author)

(Remarks on code availability)

Version 2:

Reviewer comments:

Reviewer #1

(Remarks to the Author)

(Remarks on code availability)

The authors have made a good faith effort to address criticisms. I think that their conclusions are sufficiently tempered that I favor publication. The methodological advances provide a nice template for how this type of imaging data can be used to discover new biology.

Reviewer #3

(Remarks to the Author)

I have no further comments.

(Remarks on code availability)

REVIEWER COMMENTS

Reviewer #1 (Remarks to the Author):

Habashy and colleagues present a new quantitative multi-modality imaging approach involving stable isotope tracers, SEM, NanoSIMS, and sophisticated data processing to study glucose metabolism at high spatial resolution in rodent liver in vivo. This manuscript is ostensibly a methods paper (Figures 1-3 and much of Figure 4) with some potentially interesting biological observations near the end of the manuscript related to organelle contact sites and glucose to biomass flux into organelles parsed as a function of contact site frequency. Given these two aspects of this manuscript, I will address each independently.

From the standpoint of the methodology, the execution of the imaging and data analysis is exceptional, which is not surprising given prior work by the laboratories involved in the project. Many of the elements of the methodology they advance have been demonstrated in prior publications by their groups or other groups, including examples of detecting flux of labeled glucose into glycogen, amongst other subcellular structures, with NanoSIMS. However, they have put these pieces together excellently, including sequential imaging and meticulous image registration (which is no small task). Moreover, the computational work is arguably the most innovative part of the manuscript, particularly when viewed together with the other aspects.

Thank you.

1) A minor point is that the foundations for this work are somewhat under-referenced in my opinion. The correlative SEM and NanoSIMS is an important advance particularly at the resolution and with organellar discrimination achieved by the Ellisman lab. However, there are important examples of EM with NanoSIMS dating back to the early 2000s, and examples of similar approaches used to follow labeled glucose into intracellular biomass including glycogen (e.g. PMID 25262580). The authors have also previously published similar correlative analyses. I think the current accomplishments are not diminished by a more nuanced presentation of prior similar efforts.

Thank you for pointing this out; we have revised our cited literature to include additional works where nanoSIMS was used to measure glycogen and compartment turnover (PMID 25262580 and PMID 38902423).

2) It seems like cellular data from different animals are merged. This may be reasonable if labeling is similar and normal (although they are not all truly independent). Bimodal distributions are cited (e.g. Figures 4 and S5), particularly at the early timepoint. In this instance, it would be particularly important to ensure that these distributions are bimodal within animals/cells and not just a reflection of animal specific differences in organelle morphology or labeling efficiency. In addition, it would be helpful to visualize key data such that the different mice can be distinguished.

Thanks for this comment. We repeated this experiment with n=4 mice in the 1-hour timepoint and observed that the bi-modal distribution in Fig4-5 was largely due to the labelling pattern of a mouse that had higher ^{13}C labelling than others (light purple, each dot = cell – graph shown below. Each color represents a different animal, for an n=5 total). Therefore, we have removed this animal from all downstream analyzes and the updated data in this manuscript contains a normal distribution. Moreover, we performed additional experiments and introduce spatial analysis pipelines (with nearest neighbor analysis) to analyze the organelle-organelle contact landscape during the transition from a fasted to a glucose-infused state.

3) ANOVA and t tests are used, according to the methods, but many of the distributions do not appear to be normally distributed, and so consideration of non parametric tests should be employed when appropriate.

We have revised our statistical analysis and employ non-parametric methods to analyze results where samples did not have a normal distribution.

From the standpoint of experimental design and biological conclusions, there are potentially interesting observations contained in Figure 4GH, but they are preliminary and speculative and therefore it is difficult to isolate a specific hypothesis tested or discovery that has been achieved with this exciting methodology (without substantial additional experimentation).

Thank you; to address these points we have significantly revised our manuscript to include i) additional in vivo experiments, ii) high-resolution electron tomography, iii) analysis of protein-protein interaction networks, and iv) new spatial analysis tools to define the ultrastructural, molecular, and spatiotemporal compartmentalization of glycogenesis in glucose-infused mice. These experiments show how glycogen synthesis occurs at the LD scaffold and in close association with the endoplasmic reticulum, which harbors key enzymes for glycogenesis initiation and glycogenolysis. Some of these scaffolds are maintained even after an overnight fast, which suggests that some LD-glycogen scaffolds can be relatively stable and could represent glycogenesis sites that stay “primed” for future glucose influxes. In addition, LD-glycogen scaffolds are conserved from nematodes to human hepatocytes, which supports the notion that this subcellular compartment is conserved across different cell models. Finally, we expanded our computational pipeline to define the landscape of organelle-organelle networks and how they are modulated in the transition from a fasting to a glucose-infused state.

Detailed experiments, methods, and results are outlined in our revised manuscript and highlighted in our point-by-point responses below.

1) There is insufficient consideration of fed controls in my opinion. Why did the investigators not have non-fasted controls for comparison to try to model steady state conditions?

We decided against including random fed state controls in our experiments because our goal was to develop MIMS-EM to track glycogenesis in an in vivo model where hepatocytes would be pushed towards glycogenesis. This can be observed in our dose-dependent experiments using 15mg/min/kg glucose infusions (and which matches the animal's EndoRa), where we see relatively low levels of ¹³C enrichment (Figure 3).

2) With fasting (particularly prolonged fast of 16 hours) followed by reversal of negative energy balance, there are undoubtedly dramatic changes in organelle dynamics in the liver. It is perhaps not surprising that there would be changes in organelle contact points and label flux, simply due to the shifting biomass and stochastic effects – and if it is stochastic as opposed to a regulated phenomenon, is it still of high biological significance?

Thank you for raising this point. First, we addressed this question by performing analysis of fasted vs glucose-infused mouse MIMS-EM data to find that some LD-ER-Glycogen scaffolds (which we propose as glycogenesis initiation sites) are preserved during fasting, which suggests these are relatively stable biological entities that could serve as primed-glycogenesis sites. Second, we combined our AI-image segmentation pipelines with spatial network analysis using nearest-neighbor analysis and Monte Carlo randomization tests to find that the overall organization of LDs and mitochondria is mostly stochastic throughout all conditions analyzed. Importantly, the spatial location of glycogen is not random as it is clustered around LD sites. We used the same approach to discover that the association of mitochondria with ER (as defined by our raster scan contact search tool) is not random during fasting. Therefore, we conclude that mitochondria-ER interactions are likely regulated during fasting periods, and these become random as glycogen fills the subcellular space. Notably, most of these specific mitochondria-associations carry small but significant changes in ¹³C enrichment – thus suggesting that these organelle sub-types could have different function/biology despite occurring at random.

This data is shown in Figures 6 and Figure S7.

3) Most of the comparisons that are statistically significant in Figure 4 are between 1 hour and 4 hours of labeling where significant augmentation of labeling is expected. It seems like the real differences that need to be flushed out are those between different organelle subtypes at the same timepoint. Such differences are referenced, but it was not clear from my reading of the figures/legends if those statistical analyses were performed and what the results were?

We have revised our manuscript and analyzes to factor these comparisons into account, as well as added n=4 mice at the 1hr timepoint (given that's where most of the heterogeneity was observed). There are indeed significant differences between 1h organelle subtypes, and these are shown in Figure S7G and overall fold enrichment from 1h are shown in Figure 7F.

4) Differences in progressive label uptake over time, especially in a system that is not at steady state does not necessarily implicate differential turnover rates. Without formally measuring turnover, the discussion of turnover should be reconsidered.

The manuscript has been re-written to incorporate the new experiments, and this specific discussion has been removed.

Reviewer #2 (Remarks to the Author):

The authors employed MIMS-EM to map the ¹³C distribution at subcellular level in hepatocytes of mice infused with ¹³C-U-glucose. They provided clear evidence that glucose-derived carbon can be mapped to specific organelles. When combined with machine-learning image segmentation, their approach revealed dynamic changes in spatial pattern of glucose ¹³C flux in hepatocytes. They also detected dynamic changes in inter-organelle contacts during the glucose infusion time.

(A) A main reservation about the study is insufficient new biology uncovered by the technique. Contrasting with a diabetes model would likely uncover biological insignificance of the morphological observations and increase the impact of the findings. The inability to detect specific metabolites also reduces the significance.

To address these points, we have revised and expanded our manuscript with additional experiments designed to investigate the intracellular organization pattern of hepatocytes during fasting and glucose-infused states. In addition, we analyzed the subcellular compartmentalization of LDs and glycogen using high-resolution electron tomography, which revealed the ultrastructure of LD-glycogen complexes to reveal that nascent glycogen particles are in contact and/or very close to the LD scaffold, and in the vicinity of ER and lysosomes. Using protein-protein interaction analysis, we show that this subcellular region is enriched in virtually all the enzymes involved in glycogenesis (and glycophagy) pathways, thus suggesting that the periphery of LDs could be an environment for compartmentalized/clustered enzymatic reactions. Accordingly, our tomography reconstructions demonstrate that glycogen complexes have a near-exponential growth curve, which supports the recently proposed model of glycogen as liquid droplets/phase separation organelles that trap signaling proteins¹. We also apply imaging mass spectrometry (MALDI-MS) of stable-isotope labelled glycogen (also ¹³C-labelled glucose and glucose-6-phosphate), which establishes the framework for future applications of this technique to detect a wider range of metabolites in situ. Finally, we developed a spatial analysis pipeline to reconstruct organelle-organelle networks at cell scales, and nearest neighbor analysis to determine the stochasticity (or not) of identified organelle contacts.

Other comments are as follows.

1) The proximity of glycogenesis with lipid droplets was shown for brown adipocytes. Is that true also in hepatocytes? What is the biochemical basis for such proximity?

We believe the data the reviewer is referring to is from², where glycogen promotes LD biogenesis during neonatal brown adipocyte differentiation. Yes, the data in our revised manuscript strongly supports the idea that LD-glycogen form a subcellular complex. While the exact biochemical basis for this proximity remains to be determined, our new electron tomography experiments clearly show that nascent glycogen molecules are very close and in contact with the LD outer membrane and ER compartments, and within the vicinity of lysosomes. This data explains how ER-resident proteins (Pp1 and Pygl) participate in the glycogenesis process, as well as places lysosomes within range of glycogen depots for potential degradation via autophagy. Strikingly, we found LD-glycogen complexes in nematodes and human hepatocytes, thus indicating that this subcellular organization is conserved across a wide range of species (worms, rodents, humans).

2) Figure 1D-E, which infusion rate do these panels correspond to? Please clarify infusion condition for each figure and whether similar findings are made with the different infusion rates.

3) Figure legends are missing.

We apologize for this oversight. Figures and legends have been revised to provide additional clarity to our experimental designs and are attached with the main manuscript file.

Reviewer #3 (Remarks to the Author):

The technique looks promising and relevant, but the authors should also comment on the "throughput-ness" of

the technique.

This is addressed in the “limitations of this study” section.

The claims of the author sound far-fetched and need to be toned down to reflect the reality of the advance. Generally, the writing style needs to be improved. There are many instances of grammatically and scientifically incorrect sentences that need to be rephrased. This detracts from the paper.

Corrections have been made throughout the manuscript.

Specific points:

1. What was the rationale for the pentobarbital-based method of anesthesia for tissue collection, especially considering the varied effects of anesthesia and/or euthanasia on rodent tissue metabolism? Refer to following literature: <https://www.ncbi.nlm.nih.gov/pmc/articles/PMC4319778/>

Our anesthesia and tissue collection protocol are designed to minimize animal distress and optimize for fast and efficient tissue preservation for microscopy and other analyzes. Since all animals were processed using the same timed protocol, the potential impact(s) of pentobarbital on animal metabolism are expected to be applied evenly throughout our dataset and not influence the relative comparisons and conclusions drawn from these experiments.

2. Where are the scale bars in Figure 2A-E? The authors must have forgotten

Fixed.

3. Lines 107-110, “To validate...Figure S2A).”: It is not clear what was used for GC-MS, was it hepatic tissue extract? If so, please explain the metabolite extraction protocol in methods section.

4. Figure ...: Define AFE in Figure legend (how is ppm related to AFE?)

5. Line 131-133, “¹³C-labeled glycogen stores...engulfed by glycogen over time...” : Statement not clear, what are the authors observing here: glycogen engulfed by what? The sentence refers to Figures 2A and 2G but it is not clear from the images too what is being looked at in terms of time, especially because LDs are not defined (via ROIs/staining) in these images 2A and 2G? Finally how do the authors know they are looking at glycogen.

Glycogen morphology in electron micrographs has been well-established for at least 50 years and can be easily distinguished by morphology and particle size^{3,4}.

6. Lines 133-134, “Similar...(Figure 2H-I)”: I am not sure what the authors are trying to convince us about? Physical vicinity of glycogen to LD's? Again, where is the LD in Figure 2H? This is a speculative statement at best, and the authors have essentially no proof about enzyme locations. The authors are overreaching.

This figure has been revised and now the proximity of LDs with glycogen is explored with different electron microscopy techniques: 1) MIMS-EM where ¹³C hotspots that overlay with glycogen are located next to LDs, 2) SEM and electron tomography of fasted and glucose-infused livers, respectively, showing that glycogen grows in the periphery of LDs (SEM data) and that this interface involves close contact between glycogen and LD scaffolds (Tomography), 3) with analysis of previously published FIB-SEM microscopy of mouse livers, and 4) meta-analysis of liver proteomics datasets, we show that the glycogen proteome is enriched in all glycogenesis enzymes. Therefore, the combination of these 4 different and complementary approaches convincingly maps the proximity of glycogen to LDs. The identification of the spatial location of enzymes is ongoing and will be a topic for a future manuscript.

7. Lines 143-144, “This suggested...and interactomes”: This sentence is claimed to be a hypothesis; however, this sentence is the re-statement of a very basic concept that- Glucose metabolism participates in multiple pathways other than glycogen synthesis including amino acid synthesis and fatty acid synthesis. Please change the word “hypothesis” in Line 145. Refer to: Han, HS., Kang, G., Kim, J. et al. Regulation of glucose metabolism from a liver-centric perspective. *Exp Mol Med* 48, e218 (2016).

<https://doi.org/10.1038/emm.2015.122>

Manuscript has been revised and this reference is now cited.

8. Line 152-153: What is the meaning of “morphological levels in hepatocytes”? The corresponding Figure 4C that has been referred to talks about fraction of cell area. Are the authors trying to say, “relative size of the organelles in the hepatocytes”?

9. Line 153-156: “This revealed....(Figure 4C).”: very long and confusing sentence. What is the meaning of “changes to the overall hepatocyte organelle composition”? The statement is grammatically extremely hard to follow and conceptually unjustified. What about suppression of de novo lipogenesis (triglycerides) in the liver being down-regulated and the effect on lipolysis being unchanged? This statement of lipolysis must be revised and further justified if the authors wish to make it.

Points 3-9 have been revised and the text edited for clarity and grammar where needed.

10. “Mesoscale channeling” claimed by the authors in the title can only be confirmed by a time-lapse imaging (not just two time points like 1 and 4 h) to see a particular labeled (or MS identified) metabolite getting channeled from one organelle to another. The existence of organelle contacts does not guarantee travel/metabolite transfer. ^{13}C enrichment can be due to multiple factors, not necessarily from one metabolite being channeled across organelles.

We agree. The title has been modified to better reflect the findings in our manuscript.

11. How are the ground truths generated in the SEM images? Is this purely on morphology? If so, the authors should truly show an additional method – fluorescence – to show the specificity of their annotations and DL segmentation.

Ground truths are generated by manual segmentation of SEM images using the characteristic anatomy of ER, LDs, glycogen, or mitochondria. Since our U-net models are trained specifically in SEM images, these would not be applicable to analyze fluorescence data.

12. A final comment is in the limitations section. If the method is limited to measuring $^{13}\text{C}/^{12}\text{C}$ ratios, what is the point of the MIMS and mass spectrometry in general? What is the advantage over Raman or vibrational microscopy, which has virtually zero sample prep that even has more chemical specificity and is isotope specific: see <https://onlinelibrary.wiley.com/doi/full/10.1002/jcb.30372> and <https://www.ncbi.nlm.nih.gov/pmc/articles/PMC6599680/>? The authors must address in their limitations section.

To clarify, MIMS-EM is not limited to measuring $^{13}\text{C}/^{12}\text{C}$ ratios. In previous publications^{5,6} (and in our current manuscript), we and others have shown that MIMS can quantify multiple isotopes (the “M” in MIMS stands for “multiple”), including ^{15}N , ^{13}C , ^{18}O , ^{32}S – for example. The reviewer is correct in the sense that Raman can provide chemical specificity, however it fails to identify a broad range of organelle identity, distribution, and stable isotope enrichment that is detected with MIMS-EM. To begin to address the lack of molecular identification of MIMS-EM, we provide a framework for the use of stable isotope MALDI-MS to quantify molecular patterns of stable isotope enrichment in separate tissue sections. We envision that future applications of orthogonal chemistry approaches such as Click-EM will add to the MIMS-EM toolset and provide chemical specificity signals.

Minor comments:

1. Line 156: “This increase in glycogen” can be replaced by “This increase in the size of glycogen stores”.
2. Line 157: Rephrase “loss in LDs...VLDL particles.” This part does not make much sense.
3. Line 167, “Within 4 hours...lost to increase....(Figure 4E)”: Sentence is unclear, rephrase.

These have been corrected.

Reviewer #3 (Remarks on code availability):

No code was made available.

We apologize; all relevant code has been uploaded to NPGs online code sharing tool.

Reviewer #4 (Remarks to the Author):

References cited:

- 1 Liu, Q. *et al.* Glycogen accumulation and phase separation drives liver tumor initiation. *Cell* **184**, 5559-5576.e5519 (2021). <https://doi.org/10.1016/j.cell.2021.10.001>
- 2 Mayeuf-Louchart, A. *et al.* Glycogen Dynamics Drives Lipid Droplet Biogenesis during Brown Adipocyte Differentiation. *Cell Rep* **29**, 1410-1418.e1416 (2019). <https://doi.org/10.1016/j.celrep.2019.09.073>
- 3 REVEL, J. P., NAPOLITANO, L. & FAWCETT, D. W. Identification of glycogen in electron micrographs of thin tissue sections. *J Biophys Biochem Cytol* **8**, 575-589 (1960). <https://doi.org/10.1083/jcb.8.3.575>
- 4 Prats, C., Graham, T. E. & Shearer, J. The dynamic life of the glycogen granule. *J Biol Chem* **293**, 7089-7098 (2018). <https://doi.org/10.1074/jbc.R117.802843>
- 5 Drigo, R. A. E. *et al.* Age Mosaicism across Multiple Scales in Adult Tissues. *Cell Metabolism* **30**, 343-+ (2019). <https://doi.org/10.1016/j.cmet.2019.05.010>
- 6 Steinhauser, M. L. *et al.* Multi-isotope imaging mass spectrometry quantifies stem cell division and metabolism. *Nature* **481**, 516-U131 (2012). <https://doi.org/10.1038/nature10734>

REVIEWER COMMENTS

Reviewer #1 (Remarks to the Author):

I have reviewed this new manuscript. The authors have clearly done substantial work in the interim since the index submission and overall have made an effort to respond to the initial reviews. The manuscript is undoubtedly improved, although not having the original manuscript with which to compare (and given the absence of clear designation of which textual components and figure components are new) it was somewhat difficult to fully assess.

Thanks. A revised version of the manuscript with tracked changes ON is now provided.

My opinion of the quality of the correlative imaging has not changed. It is spectacular.

Thanks!

My lingering concerns about this manuscript are two-fold:

1) The demonstration of inter-organelle contact points may be interesting, but in the current form is underdeveloped. In short, it is not clear that the methodology in this manuscript has been leveraged to make an impactful biological discovery. They may have! -- but it still seems immature from that perspective. For example, if I understand their Monte Carlo analyses, it shows that many of these interactions probably are stochastic, although this does not apply to glycogen. However, does this just suggest clustering of glycogen granules, which is already known? *We believe the reviewer is correct in this observation, in that glycogen molecules are clustered and this is what is indeed being reported by our Monte Carlo analysis. We adjusted the text to reflect this observation:*

“This analysis determined that glycogen distribution is not random and clustered (likely due to its granular landscape), whereas LDs and mitochondria are randomly distributed within the cell cytosol (Figure 5C).”

It seems like many of these interactions might be expected based on the dramatic contraction and then expansion of glycogen stores that comes with prolonged fasting followed by a glucose load, not to mention the additional potential confounding factor of intracellular trafficking of newly synthesized glycogen and therefore in its current form it is difficult to know what this really means. I remain intrigued by how the patterns would differ in a fed control state.

We have added analyses showing that the drop in mitochondrial-ER contacts happens at a timepoint before significant glycogen accumulation and in the absence of differences in cell size. We also compare these results with hepatocytes from mice in a fed state. In addition, we placed representative SEM images to illustrate the changes in mito-ER contact density detected using our raster scan method (see Figure 5H). This is now addressed in this section:

“Notably, 1 hour of glucose infusion led to a ~80% decrease in mitochondria-ER contacts (Figure 5G) despite small differences in glycogen content and no differences in cell size (Figure 3K, Figure S8F). This phenotype was sustained after 4 hours of glucose infusion and observed in random fed state hepatocytes as well (Figure 5E-H).”

2) I continue to have some concerns about the data analyses and merger of cells from different animals. Simply showing that one animal in a relatively small experiment (n=5) is contributing more to the variance than the other 4 animals in my view is not sufficient to exclude it. One could do a formal 'outlier' test, but excluding on that basis in this context would be somewhat controversial. Beyond the question of excluding that animal, cells from different animals should not simply be merged in this way because the data points cannot all be viewed as independent-- it seems to be a merger of technical replicates (hepatocytes) from different biological replicates (n=4-5 mice). At the very least, it would seem that there should be a greater effort to reveal the

variance across animals in these experiments. They get close with the color-coded dot plot shown in the response document, but no such formal analyses or data visualization seems to be present in the manuscript itself.

Thank you for raising this point. We have re-included the “outlier” animal together with an expanded dataset that relays and shows the variation at animal and cell metrics. Moreover, we believe that cell-level metrics and variation are the most representative scales for our organelle-level analyses. To show the reproducibility and variability in our data, we have included data analysis of ¹³C enrichment at cell and organelle scale (Figure 3I-J, S4C), fraction of cell area covered by organelles (Figure 3K, S4D), and mitochondria-ER contact frequencies across organelle, cell, and animal scales in Figure 5E-G.

Regarding Reviewer 2's concerns:

A main reservation about the study is insufficient new biology uncovered by the technique. This remains a concern. They have shared new data, but have not really made a central mechanistic discovery beyond showing associations of new glycogen with different intracellular compartments including lipid droplets. They have reanalyzed MS data and find additional circumstantial evidence (PPIs) to support potential physical/functional interactions between compartments, but I'm not sure demonstrating interactions between growing glycogen molecules and other cytoplasmic structures in isolation is a critical new discovery in isolation.

Contrasting with a diabetes model would likely uncover biological insignificance of the morphological observations and increase the impact of the findings.

They did not do additional models. Without looking at additional models, it is difficult to know what they would add, as this is not necessarily a request that addresses a narrow hypothesis. I agree with the original reviewer that additional data might have provided context. Whether it was the disease model or some other experimental condition that could have yielded a clearer 'discovery' I don't know, but this particular point raised by the Reviewer speaks to the first point above.

We strongly believe that adding diabetes/metabolic disease models goes beyond the scope of this original work and will be investigated during future work. Nevertheless, to provide more physiological context to our discoveries and spatial tools, we now include data from fed animals (see comment above, data shown in Figure 5E-H).

The inability to detect specific metabolites also reduces the significance. They did MALDI to show some molecular specificity. However, the MALDI does not add much, because of its low resolution. I don't know of an obvious and direct way to address this question and bring molecular specificity at the resolution of SEM or NanoSIMS. While I don't view the MALDI data as being an important addition, I also don't fully agree with the original question raised by Reviewer 2 that this is an important issue.

Thanks. We believe that our MALDI-MS experiments show significant technological advancement that is in line with the edge of the MALDI imaging field. We are confident that our studies (and supported by a recent study in glycogen MALDI-MS in lung cancer¹) will drive innovation forward and increase the application of stable isotope MALDI-MS in animal and human disease models.

Beyond these larger questions of novelty/impact, the specific points raised by Reviewer 2 seem to have been addressed, including have some data in BAT/hepatocytes.

Thanks!

Reviewer #3 (Remarks to the Author):

The authors have worked duly on the manuscript and the clarity of writing has significantly improved. This allows for a better appreciation of the technique/science being described. The addition of eTOMO and Monte Carlo simulation analysis are definitely making the paper more convincing.

Thank you.

However, there are some places with errors need to be fixed and some points that need to be clarified. These are listed below:

1. Point 3 of my comments has not been responded to in the response doc (despite the authors saying at point 9 that points 3-9 have been resolved). While it does appear that the tissue type used for GC-MS has been clarified in the results section (lines 123-124), however the methods of metabolite extraction employed for GC-MS samples has not been elucidated as requested in the review.

Thank you for pointing this out. This method section was left out of the previous version. The methods for metabolite GC-MS analysis are now included in the methods sections.

2. Also, line 125 claims Figure S2A to describe the GC-MS data which is not the case. Such errors of mis-referencing of figures are present throughout the manuscript, having a negative impact on the reader and providing a poor impression of the authors and, frankly, the journal.

We sincerely apologize for this mistake. The data in question is shown in Figure S1H. Different authors have carefully edited and re-read the manuscript (separately) to ensure that appropriate figure references and legends are in place.

3. Point no. 4 in the list of comments has not been addressed at all, neither in the response document, nor in the manuscript. Please address

This definition is now listed in the legend to Figure 1. *AFE, atomic fractional enrichment.*

4. Response to point 5 is incomplete. What is the rationale for using the word “engulf”? Is saying that “LD’s were profusely surrounded by ^{13}C glycogen stores” a better way of phrasing it?

We have changed (see **bold**) this sentence to: *“This association was underscored in ^{13}C -glucose-infused mice, where newly synthesized glycogen stores were in direct contact and/or within the immediate neighborhood of LDs within 1 hour; by 4 hours, LDs were largely surrounded by ^{13}C glycogen (Figure 4A).”*

5. Point no. 8 in the list of comments have not been addressed in the response file. It appears from the manuscript file that the figure as a whole and the corresponding text have been replaced but it would be helpful to get a direct point-by-point response from the authors.

We apologize. The text has been re-written and now track changes are ON. We do not mention “morphological features” in the manuscript; instead, we describe (throughout the text) how fasting and glucose infusion modulate organelle distribution and connectomics.

6. One of the authors in the original paper “Tom Deerinck” has not been named in the revised manuscript. Has their contribution been removed from the paper or is this a typo?

Tom was removed by mistake during copy editing of the last version. We have corrected this typo.

Reviewer #3 (Remarks on code availability):

I have briefly reviewed the code - actually the ReadMe.m file, which appears to give instructions on how to use the code.

No response required.

Reviewer #4 (Remarks to the Author):

No response required.

References cited

- 1 Clarke, H. A. *et al.* Glycogen drives tumour initiation and progression in lung adenocarcinoma. *Nat Metab* (2025). <https://doi.org/10.1038/s42255-025-01243-8>